# Feature sequence-based genome mining uncovers the hidden diversity of bacterial siderophore pathways

**Shaohua Gu[1†], Yuanzhe Shao[2†], Karoline Rehm[3], Laurent Bigler[3], Di Zhang[1], Ruolin He[1], Ruichen Xu[4], Jiqi Shao[1], Alexandre Jousset[5], Ville-Petri Friman[6], Xiaoying Bian[7], Zhong Wei[5]\*, Rolf Kümmerli[8]\*, Zhiyuan Li[1,2]\***

[1]Center for Quantitative Biology, Academy for Advanced Interdisciplinary Studies, Peking University, Beijing, China; [2]Peking-Tsinghua Center for Life Sciences, Academy for Advanced Interdisciplinary Studies, Peking University, Beijing, China; [3]University of Zurich, Department of Chemistry, Zurich, Switzerland; [4]School of Life Science, Shandong University, Qingdao, China; [5]Jiangsu Provincial Key Lab for Organic Solid Waste Utilization, Key Lab of Organic-based Fertilizers of China, Nanjing Agricultural University, Nanjing, China; [6]University of Helsinki, Department of Microbiology, Helsinki, Finland; [7]Helmholtz International Lab for Anti-infectives, State Key Laboratory of Microbial Technology, Shandong University, Qingdao, China; [8]University of Zurich, Department of Quantitative Biomedicine, Zurich, Switzerland

**\*For correspondence:**
weizhong@njau.edu.cn (ZW);
rolf.kuemmerli@uzh.ch (RK);
zhiyuanli@pku.edu.cn (ZL)

[†]These authors contributed equally to this work

**Competing interest:** The authors declare that no competing interests exist.

**Abstract** Microbial secondary metabolites are a rich source for pharmaceutical discoveries and play crucial ecological functions. While tools exist to identify secondary metabolite clusters in genomes, precise sequence-to-function mapping remains challenging because neither function nor substrate specificity of biosynthesis enzymes can accurately be predicted. Here, we developed a knowledge-guided bioinformatic pipeline to solve these issues. We analyzed 1928 genomes of *Pseudomonas* bacteria and focused on iron-scavenging pyoverdines as model metabolites. Our pipeline predicted 188 chemically different pyoverdines with nearly 100% structural accuracy and the presence of 94 distinct receptor groups required for the uptake of iron-loaded pyoverdines. Our pipeline unveils an enormous yet overlooked diversity of siderophores (151 new structures) and receptors (91 new groups). Our approach, combining feature sequence with phylogenetic approaches, is extendable to other metabolites and microbial genera, and thus emerges as powerful tool to reconstruct bacterial secondary metabolism pathways based on sequence data.

## eLife assessment

This **important** study presents a novel pipeline for the large-scale genomic prediction of members of the non-ribosomal peptide group of pyoverdines based on a dataset from nearly 2000 Pseudomonas genomes. The advance presented in this study is based on **convincing** evidence. This study of bacterial siderophores has broad theoretical and practical implications beyond a singular subfield.

## Introduction

Rapid advancements in sequencing technologies have revolutionized our view on microbial communities. While amplicon sequencing provides information on community composition and diversity, shotgun and whole-genome sequencing allow us to reliably anticipate evolutionary and ecological relationships between microbes and to obtain functional information on communities. Computational

models assessing the metabolic capacity of individual members, or an entire consortium, have become very popular and powerful (*Zengler and Palsson, 2012*; *Gu et al., 2019*; *García-Jiménez et al., 2021*). The major focus of such modeling approaches is typically on the primary metabolism of bacteria, as genes involved in core metabolic pathways are highly conserved and can be identified with relative ease (*Gu et al., 2019*; *Colarusso et al., 2021*). Conversely, analysis of the secondary metabolites has attracted less attention, even though they include compounds such as antibiotics, toxins, siderophores, biosurfactants, all known to have important implications for microbial community assembly (*Scherlach and Hertweck, 2018*; *Trivedi et al., 2020*) and to be important sources for pharmaceutical discoveries (*Price-Whelan et al., 2006*; *Thirumurugan et al., 2018*; *Chaturvedi et al., 2012*).

There are multiple challenges that currently prevent a detailed unraveling of secondary metabolism of bacteria based on genome data (*Penn et al., 2009*; *Yee et al., 2023*; *Lautru et al., 2005*). First, most secondary metabolites are produced by pathways comprised of modular enzymes such as non-ribosomal peptide synthetases (NRPSs) or polyketide synthases (PKS) (*Andryukov et al., 2019*; *Kautsar et al., 2021*). Locating complete biosynthesis clusters and identifying all enzyme-encoding genes is challenging from highly fragmented metagenomic sequences or draft genomes with a high number of contigs. Second, functional predictions for coding regions within a cluster rely on homologous comparisons with experimentally characterized genes. Such information is often restricted to a limited number of model organisms, meaning that only a small portion of the existing secondary metabolism pathways is covered by current databases. Finally, given the complex multi-modular biosynthesis machineries, it is challenging to precisely predict the secondary metabolites produced even with accurately annotated NRPS or PKS clusters. The main challenge is that a large pool of non-proteinogenic amino acids is used as substrates and the specificity of an enzyme's A domain, connecting these unusual amino acids, is often poorly understood (*He et al., 2023*). As a result, new computational methods are needed to accurately reconstruct bacterial secondary metabolism from sequence data.

Here, we present a new bioinformatic pipeline that overcomes these challenges. We specifically focus on a particular class of secondary metabolites (iron-scavenging siderophores) as a case study to develop a bioinformatic workflow that predicts the chemical structure of the produced metabolites with nearly 100% accuracy. Our pipeline is based on improved gene annotation combined with a phylogeny- and feature sequence-based substrate prediction techniques (*Figure 1*). In comparison with the currently available databases and bioinformatic tools (*Blin et al., 2023*), the main advancement of our method is the more accurate prediction of synthesized products based on non-ribosomal peptide synthetase (NRPS) clusters identified in genome data. In addition, we show that our workflow is capable of correcting previous misannotations and is extendable to other secondary metabolites (e.g. toxins, biosurfactants, virulence factors) and microbial genera.

Among siderophores, we focus on NRPS machineries that are responsible for the biosynthesis of pyoverdines, a class of chemically diverse siderophores with high iron affinity, produced by *Pseudomonas* bacteria (*Ringel and Brüser, 2018*; *Hopkinson and Morel, 2009*). While each *Pseudomonas* strain produces a single type of pyoverdine, an enormous structural diversity has been described across strains and species (*Cobessi et al., 2005*; *Diggle and Whiteley, 2020*; *Meyer et al., 1997*; *Bodilis et al., 2009*). Pyoverdine types differ in their peptide backbone, meaning that the diversity should be mirrored in NRPS enzyme diversity and their selectivity for the different amino acid substrates (*Hopkinson and Morel, 2009*). Based on this knowledge, our pipeline entails the following steps (*Figure 1*): (i) identification of the complete sequences of pyoverdine synthetase genes from fragmented draft genomes, (ii) building the pyoverdine biosynthesis machinery in silico by extracting the feature sequences for substrate specificity from motif-standardized NRPSs, and (iii) predicting the precise chemical structure of pyoverdines followed by empirical verification.

An additional element of iron acquisition involves specific receptors that recognize and trigger the uptake of iron-loaded siderophores. Pyoverdine receptors are annotated as FpvA and it is known that receptor diversity matches pyoverdine diversity (*Bodilis et al., 2009*; *Kümmerli, 2023*). Moreover, FpvA belongs to the family of TonB-dependent receptors and a single *Pseudomonas* species often has multiple gene copies encoding different receptors. This poses an additional bioinformatic challenge: how to find the gene encoding the specific pyoverdine receptor among several potential receptor genes? To overcome this, we develop an algorithm that focuses on receptor sequence regions involved in pyoverdine recognition and translocation across the outer membrane with supervised

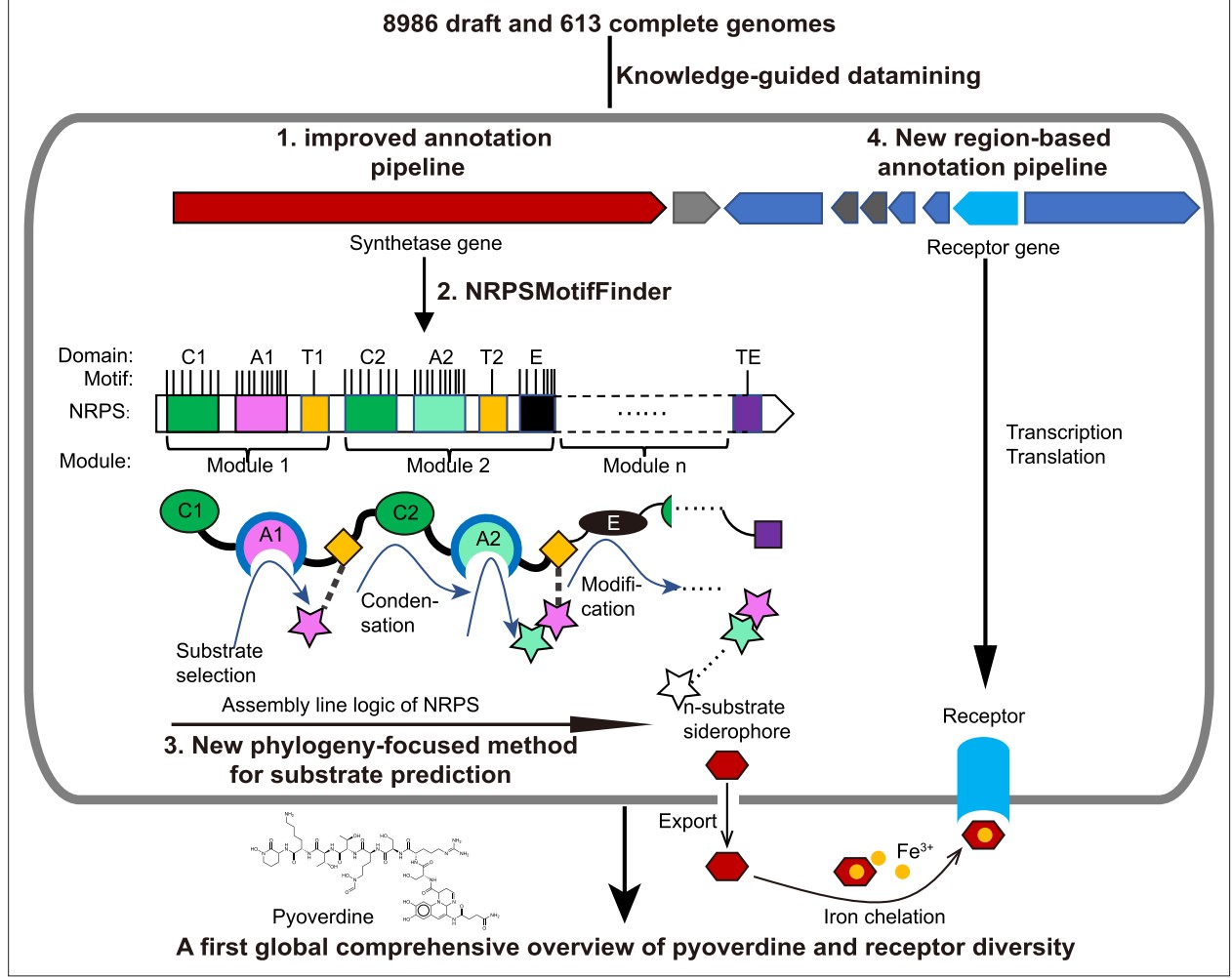

**Figure 1.** Scheme depicting our new genome mining pipeline to precisely predict the biosynthesis, the molecular structure, and the uptake machinery of pyoverdines, a family of iron-scavenging siderophores produced by members of the *Pseudomonas* genus. The gray rounded outer rectangle represents a bacterial cell. The red and blue arrow-shaped boxes stand for the synthetase and receptor genes for pyoverdines, respectively. Synthetase genes are transcribed and translated to form the *n*-modular non-ribosomal peptide synthetase (NRPS) enzymes. These enzymes synthesize the peptide backbone of pyoverdine through an assembly line using their repeating module units, with the A domain being responsible for substrate selection and the E domain for chirality. The *n*-substrate siderophores are then exported to the extracellular space for iron chelation. Membrane-embedded TonB-dependent receptors recognize the ferri-siderophore complex and import it into the cell. Bold black text and black arrows describe our multi-step computational methods developed to reconstruct the entire process from genome sequence data. First, the annotation pipeline was improved (from antiSMASH, *Blin et al., 2019*) to extract the complete sequence of pyoverdine synthetase genes from draft genomes. Second, NRPSMotifFinder was used to define A and E domains and to determine the exact motif-intermotif structure of the pyoverdine assembly line. Third, intermotif regions most indicative of substrate specificity were used to develop a phylogeny-focused method for precise product prediction. Fourth, a sequence region-based annotation method was combined with genome architecture features to identify the FpvA, receptors responsible for ferri-pyoverdine import.

learning methods that reliably locate *fpvA* genes in fragmented genomes based on these regions (*Figure 1*). Not only did our method unveil a substantial number of previously unrecognized pyoverdine receptors, but it also revealed that certain receptor groups are widely distributed among *Pseudomonas* strains, opening the possibility of pyoverdine sharing and exploitation across the strain and species boundaries. Altogether, our bioinformatic pipeline uses knowledge-guided insights empowered by supervised learning to construct a first systematic sequence-to-function mapping of a family of secondary metabolites (pyoverdine) and their corresponding receptors. Our analysis unveils a yet unrecognized extraordinary diversity of iron-scavenging machineries in pseudomonads.

# Results

## Section 1: Improved annotation pipeline reveals a vast reservoir of pyoverdine synthetase genes

The first step of our bioinformatic pipeline was to improve the annotation of pyoverdine synthetase genes. The pyoverdine molecules is composed of a conserved fluorescent chromophore (Flu) and a peptide chain (Pep), which are both synthesized by NRPS enzymes (*Meyer, 2000*). There exist already tools, such as antiSMASH, that can find and annotate NRPS clusters in microbial genomes (*Blin et al., 2019*). However, antiSMASH (and other popular annotation platforms; *Xu et al., 2023*; *Keller, 2019*) rely on accurate gene predictions, which are typically problematic for fragmented genomes. Consequently, while antiSMASH can recognize and annotate certain genes of an NRPS cluster, the precise reconstruction of a complete NRPS assembly line often fails. This is particularly problematic because most available genomes are drafts and any analysis suffers from the unavoidable issue of incomplete or misannotation of gene fragments.

To overcome these issues, we developed an improved four-step annotation pipeline starting with the raw annotation of the pyoverdine cluster obtained from antiSMASH (*Blin et al., 2019*; *Figure 2a*). First, we implemented an NRPS hidden Markov model (HMM) to re-annotate and extract the entire nucleotide sequence of the pyoverdine synthetase cluster (*Bateman et al., 2004*), including the genes missed by antiSMASH. For this step, the nucleotide sequences were converted into amino acid sequences to avoid erroneous gene predictions typically associated with antiSMASH. Second, we assembled the entire re-annotated pyoverdine coding region into a single sequence with defined start (co-enzyme A ligase) and end (thioesterase, TE) markers. In most cases (~90%), there are two separate biosynthetic gene clusters for Flu and Pep. Their ordering is easy as Flu always precedes Pep. In the remaining cases (~10%), there is either one or three biosynthetic gene clusters. No ordering is required for the former. For the latter, we put Flu first and the cluster with TE domain at the end. Third, we used NRPSMotifFinder to identify the C, A, T, E, and TE motifs that determine the NRPS structure of pyoverdine (*He et al., 2023*). Finally, we applied a safety measure to ensure that the recovered NRPS assembly line is complete (contains both Flu and Pep) and is not truncated, which can occur when a synthetase coding region is at the edge of a contig. Consequently, we dismissed all pyoverdine biosynthesis clusters located within 100 bp proximity to contigs' edges and either lacked Flu or Pep synthetic genes.

Next, we applied our improved pyoverdine biosynthesis annotation pipeline to 9599 *Pseudomonas* genomes (including 613 complete and 8986 draft genomes) retrieved from the Pseudomonas Genome Database (*Winsor et al., 2016*). We found the pyoverdine biosynthesis machinery in 97% of the genomes (*Figure 2b*), indicating that the machinery is ubiquitous in *Pseudomonas*. However, since 94% of the analyzed genomes were in draft form, the pyoverdine biosynthesis machinery was likely truncated (i.e. on the edge of the contig) in 63.4% (6087) of the genomes. These genomes were excluded from further analysis. Around 3.1% of retained genomes (293) with high assembly completeness were missing pyoverdine synthetic genes, indicating that these *Pseudomonas* strains were not able to produce pyoverdine ('non-producers'). The rest of the genomes (33.5%; 3219 genomes) were classified as 'producers' with complete pyoverdine NRPS assembly lines that meet all our quality controls. For these 3219 genomes, we used NRPSMotifFinder to find boundaries between the various biosynthesis domains and to determine the amino acid length of the cluster and the number of A domains. The lengths of pyoverdine synthetic clusters ranged between 7690 and 21,333 amino acids, and the number of A domains per synthetase ranged between 6 and 17, with a total of 35,281 A domains being present across all strains (*Figure 2c*). Overall, our analysis pipeline unveiled a vast diversity of pyoverdine synthetases that goes far beyond of what has previously been described in the literature.

Finally, we utilized the PhyloPhlAn3 pipeline (*Asnicar et al., 2020*) to conduct a phylogenetic analysis based on 400 conserved genes with the 293 non-producers and the 3219 producers. We first removed redundant non-producers by retaining the most integrative genome among strains with high phylogenic similarity (>95% identity). Then, we removed redundant producers by retaining the most integrative genome among strains with high phylogenic and pyoverdine synthetase similarity (*Figure 2d*, >95% identity). This data cleaning yielded a total of 1928 *Pseudomonas* strains (403 complete and 1525 incomplete genomes), segregating into 1664 pyoverdine producers and 264 non-producers. The phylogenetic tree revealed that all major *Pseudomonas* species clades were present in

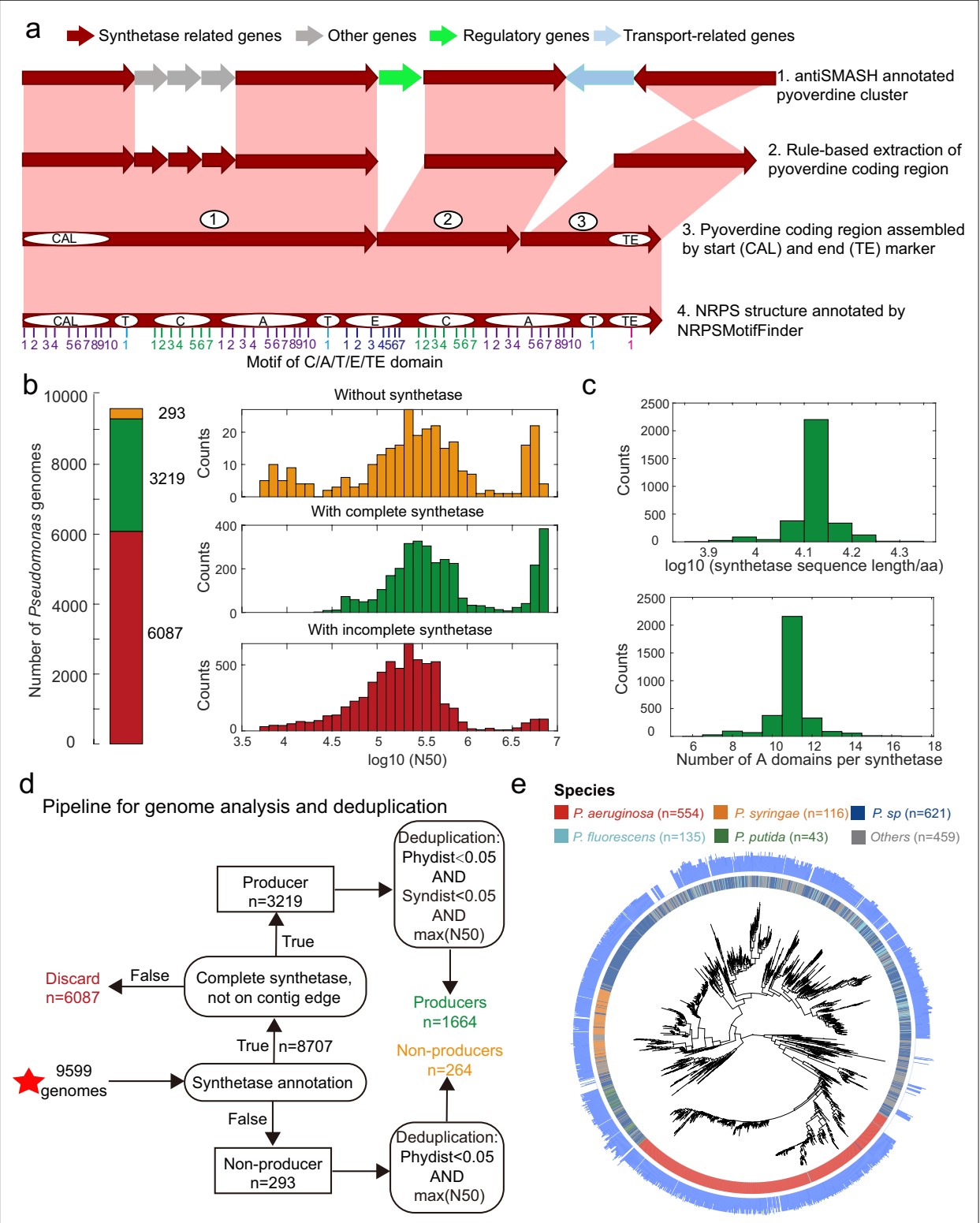

**Figure 2.** Improved annotation pipeline reveals a vast diversity of pyoverdine synthetase genes. (**a**) Improved annotation pipeline based on the raw annotation from antiSMASH (***Blin et al., 2019***). (**b**) The annotation pipeline was applied to 9599 *Pseudomonas* genomes (94% draft genomes). Genomes could be separated into three categories. Yellow: genomes without pyoverdine cluster. Green: genomes with a complete pyoverdine cluster. Red: genomes with incomplete pyoverdine synthetase cluster. The red category involved genomes with truly incomplete clusters (lacking Flu or Pep synthetic genes) or genomes with likely truncated synthetic genes at the edge of contigs. (**c**) Distributions of the sequence length (upper panel) and the number of A domains (lower panel) across all the genomes with a complete synthetase cluster. (**d**) Workflow applied to separate the 9599 *Pseudomonas*

*Figure 2 continued on next page*

*Figure 2 continued*

genomes into the three categories described in b and removing of redundant genomes with high phylogenic similarity and showing high similarity in pyoverdine synthetases. Red star indicates the start of the workflow. (**e**) Phylogenetic tree depicting the relationship among the 1928 non-redundant *Pseudomonas* strains (1664 producers and 264 non-producers) based on the concatenated alignment of 400 single-copy conserved genes in their genomes. The inner ring depicts the taxonomical classification including the four most prevalent species. The outer ring shows the number of A domains present in the pyoverdine synthetase assembly line in each strain.

our dataset (*Figure 2e*). Moreover, the number of A domains varied widely among species and even between strains within species. For example, the number of *Pseudomonas aeruginosa* A domains ranges between 7 and 14. In summary, by improving the synthetase annotation method, we successfully obtained 1664 highly reliable pyoverdine synthetases (with a total of 18,292 A domains) and 264 non-producers.

## Section 2: Phylogeny-focused substrate prediction for pyoverdine A domains

Our next goal was to precisely predict the molecular structure of the pyoverdines produced by the 1664 strains with complete synthetase gene clusters. The first essential step toward this goal was to reliably predict the substrate selectivity of all A domains in the NRPS assembly line. The A domain of each module selects for a single substrate among 22 proteinogenic and hundreds of non-proteinogenic amino acids (*Felnagle et al., 2008*; *Süssmuth and Mainz, 2017*). Moreover, whenever an E domain exists downstream of an A domain, the chirality of the amino acid incorporated into the peptide chain gets modified from L to D. Thus, the modularity combined with the selectivity of A domains can promote an enormous diversity of pyoverdine molecule structures. To date, at least 73 pyoverdine structures have been reported (*Supplementary file 1*). Among those, we could identify 13 cases for which both structures (obtained from structural analysis experiments, we wanted to avoid any form of biases that bioinformatic pre-predictions could introduce to downstream analyses) and completely sequenced synthetase genes were available (*Supplementary file 2*). In order to make reliable molecule predictions, two challenges must be addressed: (i) the extraction of relevant information from A domain sequences for which the substrate is known, and (ii) the effective application of this information to predict specificity of A domain sequences for which the substrate is unknown.

To address the first challenge, we built our analysis on the NRPS assembly lines of the 13 known pyoverdines identified above to extract relevant information from the A domain sequences. From this dataset, we could identify 101 A domains that are linked to 13 experimentally confirmed amino acid substrates (*Supplementary file 3*). We next performed a multisequence alignment of the 101 A domains to determine (i) the 'feature sequence', defined as the most informative sequence for substrate selectivity and (ii) the 'feature sequence distance', which allows to reliably differentiate between two different substrates. To this end, we tested three different A domain regions, three different sequence similarity measurements, and seven different clustering methods for their predictive power (*Figure 3a*). We found that the full A domain sequence is not informative for substrate prediction (*Figure 3b*, left panel). Instead, our analysis indicated that information-rich positions start with motif A4 and end before motif A5, consistent with the known role of the A domain pocket in substrate selectivity (*He et al., 2023*). Overall, the sequence region from motifs A4 to A5 (termed 'Amotif4-5'), in conjunction with Jukes-Cantor distance and Ward linkage clustering, performed best in accurately distinguishing between different substrates and maintaining homogeneity for identical substrates (*Figure 3b*). Utilizing these parameters, we conducted a comprehensive analysis of all A domains and their respective substrate information within the *Pseudomonas* NRPS biosynthetic gene cluster available in the MIBiG database (*Terlouw et al., 2023*; *Supplementary file 4*). To our delight, our method can find and correct previously misclassified substrate information of A domains in the MIBiG database (*Figure 3—figure supplement 1* and *Supplementary file 5*). This demonstrates the effectiveness of performing substrate predictions for the A domain based on feature sequences and the selected parameter combination.

To address the second challenge, we developed a 'phylogeny-focused method' to apply the above-derived feature sequence distance approach to the 18,292 discovered A domains. We realized that a direct construction of a phylogenetic tree including all 18,292 query A domains and the 101 reference A domains would be computationally too demanding and impossible to scale up. Furthermore,

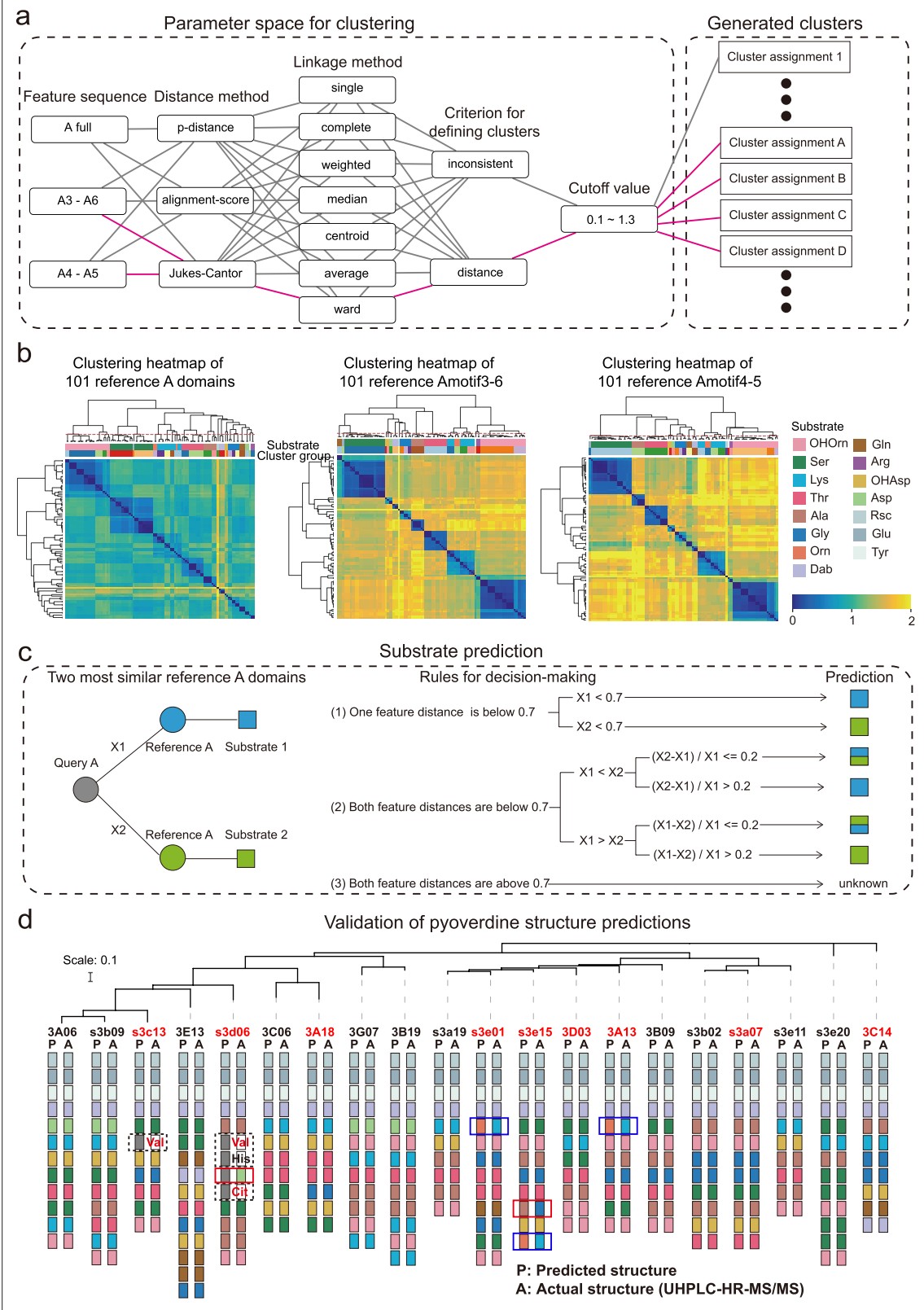

**Figure 3.** Phylogeny-focused substrate prediction for pyoverdine synthetase assembly lines. (**a**) Information from 101 reference A domains with known amino acid substrates were used to develop an algorithm that predicts substrates from A domain sequence data with high accuracy. The challenge is to group the variable A domains into clusters that predict the same substrate (captured by the silhouette index). To find the most distinctive algorithm, we combined different feature sequences of A domains (Amotif) with different distance and linkage methods in our hierarchical clustering analyses.

*Figure 3 continued on next page*

*Figure 3 continued*

The best performing path is shown in pink. (**b**) Heatmap showing the hierarchically clustered distances of the 101 reference A domains as a function of the feature sequence used. Color bar in the lower right shows the sequence distance. Left panel: complete A domain sequences. Middle panel: Amotif3-6 sequences. Right panel: Amotif4-5 sequences. The heatmaps show that hierarchical clustering, reliably associating sequence distances with substrates, worked best with the Amotif4-5. The experimentally validated substrates are shown on top of each heatmap and consist of 13 amino acids and 2 amino acid derivatives: Ser (serine), Lys (lysine), Thr (threonine), Ala (alanine), Gly (glycine), Gln (glutamine), Arg (arginine), Asp (aspartic acid), Glu (glutamic acid), Tyr (tyrosine), Orn (ornithine), Rsc (succinic acid derivatives), Dab (2,4-diaminobutyric acid), OHAsp (aspartic acid derivative), and OHOrn (a general term for three ornithine derivatives: FoOHOrn, AcOHOrn, cOHOrn). (**c**) Phylogeny-focused substrate prediction pipeline for query A domains (gray circle) based on Amotif4-5 feature sequence comparisons. X1 and X2 represent the feature distance between the query A domain and two closest reference A domains (blue and green circles), respectively. Three rules are used, based on the feature sequence distances X1 and X2 and a threshold value of 0.7 (50% similarity), to make substrate predictions for the query A domain. There are three possible outcomes: unambiguous substrate prediction (blue or green squares), ambiguous substrate prediction (dual-colored squares), and no prediction ('unknown'). (**d**) Phylogenetic tree of 20 *Pseudomonas* strains and visualization of their predicted and actual pyoverdine structures to validate our phylogeny-focused substrate prediction pipeline. Strains marked in red font indicate cases with novel (not yet characterized) pyoverdines structures. 151 out of the 160 substrates (94.4%) were correctly predicted. The nine inconsistencies are boxed in blue (lysine and ornithine are indistinguishable), in dashed black (detection of 'unknown' substrates), and in red (true mismatches). Note that our prediction pipeline (as any other pipeline) cannot distinguish between modified variants of the same amino acid, for example, we cannot distinguish the three derivatives of ornithine (FoOHOrn, AcOHOrn, cOHOrn).

The online version of this article includes the following figure supplement(s) for figure 3:

**Figure supplement 1.** Clustering heatmap of sequence distance matrix between 318 Amotif4-5 from *Pseudomonas* strains in the MIBiG database (*Supplementary file 4*).

**Figure supplement 2.** Comparison of prediction accuracy in the structural composition of secondary metabolites between our approach and antiSMASH in *Burkholderiales*.

such an approach would result in phylogeny-interference issues, where domains would cluster not only based on their substrate similarities but also based on overall species relatedness (*He et al., 2023*). To minimize the effect of phylogeny and speed up calculations, we took each of the 18,292 query A domains and identified the two most similar A domain clusters among our 101 reference A domains. We then compared the feature distance between each query A domain and the two most similar reference A domains and assigned a substrate to the query A domain using the following rules (*Figure 3c*). (1) If the feature sequence distance is below the 0.7 threshold (corresponding to 50% identity) for only one of the two reference A domains, then the substrate of the query A domain is matched to the substrate of the more similar (lower distance) reference A domain. (2a) If the feature sequence distance is below the 0.7 threshold for both reference A domains, then we considered the relative difference of the query A domain toward the two reference A domains. If the relative difference is larger than 0.2, the query A domain is matched to the substrate of the more similar reference A domain. (2b) If the relative difference is smaller than 0.2, the substrate of the query A domain cannot unambiguously be determined and is thus matched with both reference substrates. (3) If the feature distance is above the 0.7 threshold (below 50% identity) for both reference A domains, then the substrate of the query A domain is marked as 'unknown'. For most query A domains, rule (1) could be applied (17,880 cases), whereas rules (2) and (3) had to be used rarely (133 and 279 cases, respectively). We applied our methodology termed 'phylogeny-focused method' to all following substrate and pyoverdine structure predictions.

## Section 3: Experimental validation of the annotation and prediction pipeline

To test whether our bioinformatic pipeline can reliably predict pyoverdine structures, we sequenced the genomes of 20 pyoverdine-producing *Pseudomonas* strains previously isolated from soil and water (*Butaitė et al., 2018*). We applied our annotation and prediction pipeline to generate predicted pyoverdine structures for all 20 strains harboring a total of 237 A domains. Notably, none of the predicted structures matched any of the 13 reference pyoverdine structures, and 9 out of the 20 structures were predicted to be novel (not yet characterized) pyoverdines. We then used the UHPLC-HR-MS/MS method developed in 2022 (*Rehm et al., 2022*) to elucidate the chemical structures of these pyoverdines. Six out of the 20 structures had already been elucidated in our previous work (*Rehm et al., 2022*), while we here used the same method to elucidate the chemical structure of the remaining 14 pyoverdines (see *Supplementary file 13* for MS/MS profiles).

We found a near-perfect match (94.4%) between the predicted and the observed pyoverdine chemical structures and were able to accurately assign amino acids in 151 out of 160 cases (*Figure 3d*). Our method demonstrated a substantial improvement compared to commonly used approaches or platforms integrating several methods (*Supplementary file 6*), such as antiSMASH 7.0 (*Blin et al., 2023*), which could only accurately assign 94 out of 160 amino acids (accuracy: 58.8%). The nine non-matching cases in our analysis segregated into three groups. In three cases (1.9%), our algorithm could not distinguish between the substrates lysine and ornithine, as these two amino acids are highly similar in terms of both their chemical structures and corresponding A domain sequences. This is the only sensitivity issue that is associated with our approach. In four cases (2.5%), our technique assigned an 'unknown' substrate to amino acids that turned out to be valine, citrulline, and histidine. Indeed, these three amino acids were not yet present in our reference dataset and therefore no correct prediction was possible. While citrulline has never been reported in pyoverdines before, valine has been found in a pyoverdine structure but no sequence data from the corresponding strain was available. These cases show that our analysis pipeline can be used to highlight new substrates. Once experimentally verified, the new A domains and their substrates can expand the reference dataset, allowing targeted improvement of our phylogeny-focused prediction technique. Finally, there were only two cases (0.8%) that represented true mismatches between observed and predicted amino acids. Altogether, our phylogeny-focused method is highly accurate in predicting pyoverdine peptide structures and in identifying unknown substrates in *Pseudomonas*.

To assess the extendibility of our pipeline, we chose *Burkholderiales* as a test case, leveraging a manually curated dataset that we built up from the literature (see Methods for details). This dataset contains 203 A domain sequences and corresponding substrate information for 34 NRPS metabolites (e.g. toxins, biosurfactants, virulence factors) across 7 genera of *Burkholderiales* (*Supplementary file 7*). We further integrated experimentally validated biosynthetic gene cluster data from *Burkholderiales* in MIBiG 3.0 (*Terlouw et al., 2023*; *Supplementary file 8*) into our manual dataset leading to a total of 302 A domains. We then split the available data into a training set (124 A domains, *Supplementary file 12*) and a test set (178 A domains, *Supplementary file 9*). As for the pyoverdines, we first trained our algorithm on the training dataset and then applied it to the test dataset. We observed that our pipeline consistently yields high prediction accuracy within *Burkholderiales* (83%), which is considerably higher than the accuracy obtained with antiSMASH 7.0 (*Blin et al., 2023*, 67%, *Figure 3—figure supplement 2* and *Supplementary file 9*). This outcome underscores the robustness of our developed prediction algorithm, highlighting its potential extension to other NRPS secondary metabolites and microbial genera.

## Section 4: Application of the annotation and prediction pipelines to a full dataset

After successful validation, we applied our bioinformatic pipeline to the 1664 complete NRPS assembly lines annotated in our genome analysis (*Figure 4* and *Figure 4—figure supplement 1*). Across all assembly lines, we were able to predict the substrates of 17,880 A domains (97.75%) without ambiguity, whereas 133 A domains (0.73%) were associated with two different substrates, and 279 A domains (1.52%) predicted an unknown substrate (similar to the cases of valine and citrulline above). After considering the presence/absence of an E domain in each module, we derived the structures of 1664 pyoverdines according to our methods in Section 2 (Phylogeny-focused substrate prediction for pyoverdine A domains).

Our prediction yielded 188 different pyoverdine molecules, out of which only 37 structures had been previously reported. While these 37 reported structures were highly abundant across strains (1103 out of 1664), our pipeline was powerful in identifying many of the rarer pyoverdine variants. Agreeing with previous studies, we observed that the fluorophore is highly conserved among the 188 predicted structures. Moreover, our analysis found that 13 amino acids form the main structural substrates of all the 188 pyoverdine structures, with most of the variation being attributable to different substrate combinations, peptide lengths, and substrate chirality (*Figure 4*).

Notably, pyoverdine structural diversity was not strongly linked to phylogeny because the same pyoverdine structure could be found in completely unrelated species, while closely related species often had different pyoverdine structures (*Figure 4*). These observations suggest that there may be both frequent recombination and horizontal gene transfer of pyoverdine synthetase clusters between

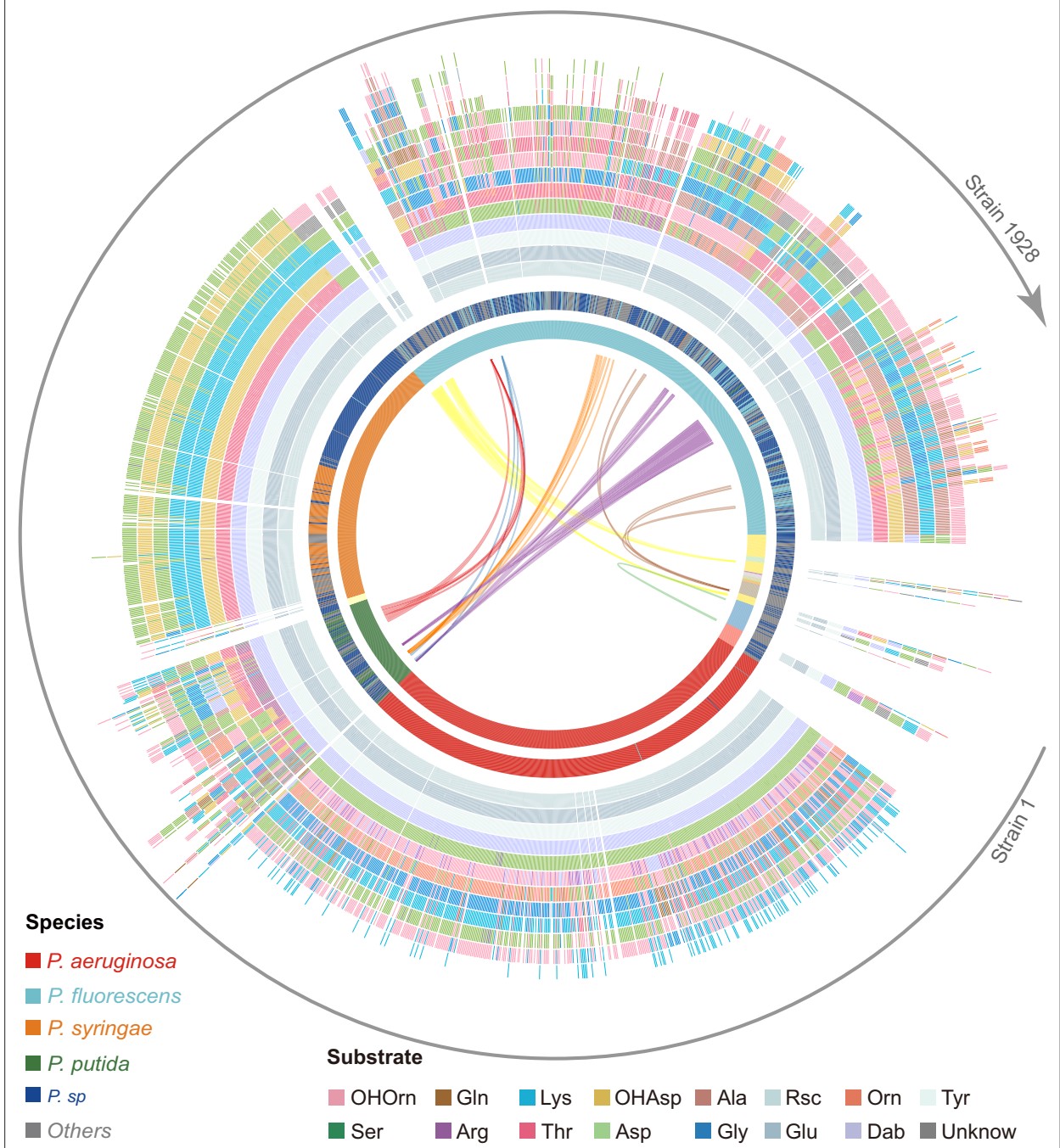

**Figure 4.** Predicted pyoverdine structural diversity based on our developed algorithm mapped onto the phylogenetic tree of all 1928 (non-redundant) *Pseudomonas* strains. In this figure, the strains are arranged based on the phylogenetic tree without showing the tree itself. The stacked boxes in the outermost circle show the predicted structure of pyoverdines, whereby each color represents a specific amino acid substrate. Strains without boxes represent non-producers (*n*=264). Boxes with two colors indicate cases of ambiguous (dual) substrate prediction. The inner circle shows the taxonomic species classification following *Figure 2e*. Because the allocation of strains to species names is often imprecise, we divided the 1928 strains by their phylogenetic distance into 18 clades (color shadings in innermost circle), out of which 13 contained more than one strain. Lines within the innermost circle link strains from different clades that share the same pyoverdine structures, whereby line colors represent the shared unique pyoverdines. The bending of the lines represents the phylogenetic sequence distances of the connected strain pairs.

The online version of this article includes the following figure supplement(s) for figure 4:

**Figure supplement 1.** Clustering heatmap of sequence distance matrix between A domains from 1664 strains and 101 reference A domains.

species. Taken together, the bioinformatics methods developed in our study can predict a suit of secondary metabolites (pyoverdines) from sequence data with high accuracy, revealing an unprecedented richness and evolutionary history of siderophores within pseudomonads and the discovery of 151 novel pyoverdine candidate variants.

## Section 5: Development of a region-based identification method for annotation of the FpvA receptors

In pseudomonads, iron-loaded pyoverdines are recognized by FpvA, a TonB-dependent receptor, that transports the ferri-pyoverdine into the periplasm (*Cobessi et al., 2005*; *Butaitè et al., 2017*; *Kramer et al., 2020*; *Jin et al., 2018*). The protein structure of characterized FpvA variants consists of three domains: The <u>S</u>ecretin and <u>T</u>onB <u>N</u>-terminus short domain (STN), the Plug domain (Plug), and the TonB-dependent receptor domain (TonB) (*Cobessi et al., 2005*). While these domains are conserved across FpvA variants and other siderophore receptors, there is substantial variation at the sequence level. This makes it challenging to reliably identify FpvA receptors from sequence data by homologous search. As an example, we were unable to find FpvA genes (with a 60% identity threshold) by homologous search in several genomes although they had complete pyoverdine biosynthesis machineries. Moreover, there are many other TonB-dependent receptors with fairly high sequence identity to FpvA but that transport other siderophores than pyoverdine (e.g. FpvB, 55% identity, transporting pyoverdine, ferrichrome, and ferrioxamine B [*Chan and Burrows, 2022*]). Therefore, it is imperative to develop a new comprehensive method for identifying FpvA receptors in *Pseudomonas* genomes.

We started our approach by comparing the sequences of 35 reported siderophore receptors, including 21 FpvA and 6 FpvB from the Pseudomonas Genome Database (https://www.pseudomonas.com/), and 8 TonB-dependent siderophore receptor sequences often found in *Pseudomonas* genomes from Uniprot (https://www.uniprot.org/), encoding receptors for the uptake of heterologous siderophores (*Supplementary file 10*). We found that all receptor sequences share a similar length of around 800 amino acids (FpvA and FpvB sequences: 809±10 amino acids). We then used the complete sequences to calculate the pairwise distances by global alignment before applying hierarchical clustering (*Figure 5a*). We found substantial divergence between FpvA variants to an extent that was comparable to the distance between FpvA and other siderophore receptors. Moreover, FpvB variants clustered with FpvA variants, showing that FpvA identification based on full sequence distances is unachievable. We hence focused on the three typical receptor domains (TonB, Plug, and STN, retrieved from the Pfam database) and applied profile hidden Markov models (pHMM) to each reference sequence to calculate the pHMM probability scores for each domain. The probability scores (calculated as the log-odd ratios for emission probabilities and log probabilities for state transitions) had reasonably high scores but no distinction was apparent between the three receptor classes (*Figure 5b*).

We next asked whether there are specific regions within the receptor sequences that are characteristic of FpvA. To address this, we conducted a multiple sequence alignment (MSA) with all 35 reference receptor sequences and mapped them onto the sequence of the well-characterized FpvA of *P. aeruginosa* PAO1 (*Figure 5c*). MSA allows to identify conserved sites (*Figure 5c*, black dots representing the top 10% most conserved sites) that are shared by the majority of the reference sequences. We then used these conserved sites to partition the MSA into variable regions which were flanked by two conserved sites. For each variable region, we assessed its predictive power to differentiate FpvA from non-FpvA sequences. For this we defined the 'FpvA identification score' analogous to the intercluster-vs-intracluster Calinski-Harabasz variance ratio, as

$$I_{FpvA} = d_{A:\,non}/d_{A:\,A}$$

where $d_{A:\,A}$ is the sequence distance among all 21 FpvA sequences, and $d_{A:\,non}$ is the sequence distance between all 21 FpvA and the 14 non-FpvA sequences.

Our analysis yielded two locations with noticeably high FpvA identification scores (*Figure 5c*). The region with the highest FpvA identification score (referred to as R1) locates at the intersection of the Plug domain and the barrel structure of the TonB domain (*Figure 5d*, between 258 Gly and 309 Gly in the PAO1 FpvA). According to the sequence distance matrix, the R1 region allows to distinguish heterologous siderophore receptors from FpvA and FpvB receptors (*Figure 5e*). The region with the second highest FpvA identification score (referred to as R2) was located in the C-terminal signaling

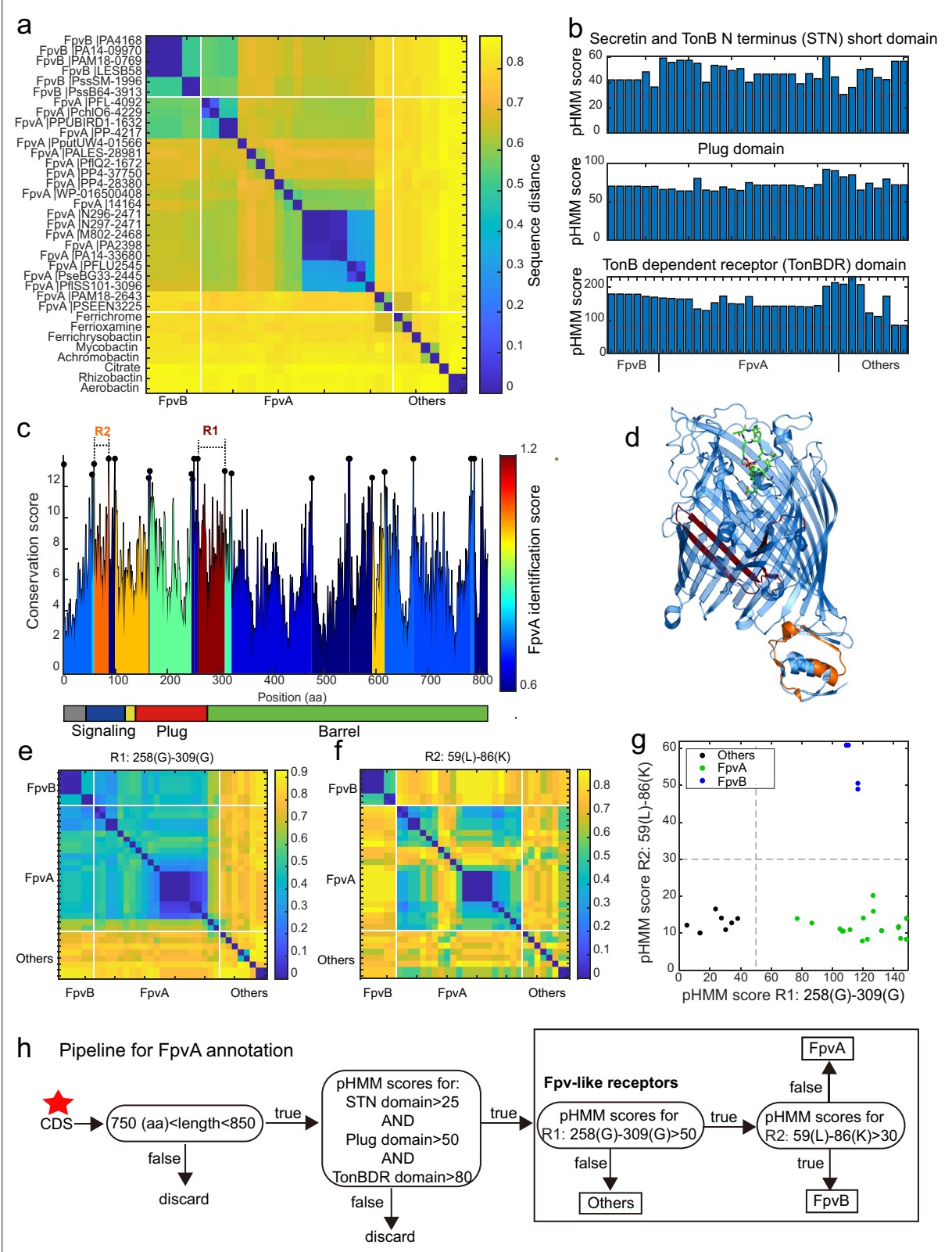

**Figure 5.** A sequence region-based identification pipeline for annotating FpvA receptors. (**a**) Heatmap displaying the hierarchically clustered sequence distances (p-distance calculation method, identity (%) = (1-sequence distance) * 100) of 35 reference siderophore receptors identified in *Pseudomonas* spp., based on full sequences. No clear discrimination between FpvA, FpvB, and other receptors is possible. The order of receptors is consistent across panels (**b**), (**e**), and (**f**). (**b**) The profile hidden Markov model (pHMM) scores of the three standard receptor domains (STN, Plug, and TonBDR)

*Figure 5 continued on next page*

*Figure 5 continued*

vary across the 35 reference sequences (A: FpvA, B: FpvB, and NA: others), but do not allow to distinguish between receptor groups. (**c**) FpvA region-based conservation scores from a multi-alignment of the 35 reference sequences mapped to the FpvA sequence of strain *P. aeruginosa* PAO1. All residues within the top 10% of the conservation score are denoted with black dots. For each region flanked by two black dots, we calculated the FpvA identification score (heatmap), representing the ability to distinguish FpvA from non-FpvA receptors. (**d**) Mapping of the two regions with the highest FpvA identification scores R1 (dark red) and R2 (orange) to the crystal structure of FpvA from PAO1 conjugated with pyoverdine (PDB 2IAH). (**e**) Heatmap showing the hierarchically clustered sequence distances of 35 reference siderophore receptors based on the R1 sequence region. A clear discrimination between FpvA/FpvB and other receptors emerges. (**f**) Heatmap showing the hierarchically clustered sequence distances of 35 reference siderophore receptors based on the R2 sequence region. A clear discrimination between FpvA and FpvB receptors emerges. (**g**) The pHMM scores of regions R1 and R2 for the 35 siderophore reference receptors are contrasted against each other, yielding a clear separation between FpvA, FpvB, and other receptors. Dashed lines indicate the pHMM threshold scores used for later analysis. (**f**) Flowchart showing all steps involved in the FpvA annotation from genome sequence data. The red star indicates the start of the workflow.

domain (*Figure 5d*, between 59 Leu and 86 Lys in the PAO1 FpvA). The sequence distance matrix revealed that R2 allows to distinguish FpvB from FpvA receptors (*Figure 5f*).

We then defined two informative pHMM scores based on (i) the alignment of the 21 FpvA sequences in the R1 region, termed R1(FpvA), and (ii) the alignment of the 6 FpvB sequences in the R2 region, termed R2(FpvB). Running R1(FpvA) and R2(FpvB) against all 35 reference sequences revealed a clear separation between the three receptor categories (*Figure 5g*). Along the R1(FpvA) axis, FpvA and FpvB reference sequences have high R1 scores (minimal score 77.0) that separate them from other siderophore receptors (maximal score 38.1), whereas FpvAs references have substantially lower R2 scores (maximal 20.2) than FpvBs (minimal 49.0) along the R2(FpvB) axis.

Based on these insights, we developed a decision flowchart for annotating FpvAs in *Pseudomonas* genomes (*Figure 5h*). First, we considered sequences as Fpv-like receptors that share similar properties to the ones identified in our reference database. Particularly, protein coding sequence (CDS) length had to be between 750 and 850 amino acids and the pHMM scores for the three typical receptor domains STN, Plug, and TonB must be greater than 25, 50, and 80, respectively (*Figure 5b*, red dashed lines). Second, we used the pHMM threshold scores obtained for R1(FpvA) and R2(FpvB) (*Figure 5g*) to differentiate other siderophore receptors (R1(FpvA) score <50) from FpvB receptors (R1(FpvA) score >50 and R2(FpvB) score >30) and FpvA receptors (R1(FpvA) score >50 and R2(FpvB) score <30). Our method effectively identifies FpvA receptors from sequence data and can be readily applied to the entire *Pseudomonas* dataset.

## Section 6: Application of the receptor annotation pipeline to the full dataset

The region-based receptor identification pipeline was applied to all 1928 *Pseudomonas* genomes. The analysis identified 4547 FpvAs, 615 FpvBs, and 9139 other TonB-dependent Fpv-like receptors across the dataset (*Figure 6a*). The 4547 FpvA sequences clustered hierarchically into 114 groups, defined by an identity threshold of 60%. When comparing to the 21 reference FpvAs (*Figure 6b*), we found that 2293 FpvA sequences have close homologues in the reference database, while 2254 FpvA sequences lack such close homologues (sequence identity <50%). These latter sequences, termed as 'dissimilar to reference', may represent novel subtypes of FpvA receptors that could not be found by simple homology search. Our analysis further shows that many strains have more than one FpvA receptor.

We then asked whether the 4547 FpvAs are found in proximity of pyoverdine Pep synthetase genes as it is commonly the case for cognate FpvA receptors (*González et al., 2021*). We thus calculated the proximity between pyoverdine Pep genes and the Fpv-like receptor genes by counting the number of base pairs between the two coding regions. All TonB-dependent receptors that have not been classified as FpvAs were more than 20 kb away from the Pep genes (*Figure 6c*). In contrast, 92% of the nearest FpvA genes were indeed located within 20 kb of their pyoverdine Pep genes (*Figure 6d*, called proximate receptors). These proximate receptors encompassed both those with close (66%) and more distant (34%) resemblance to the reference receptor types. Overall, this proximity analysis confirmed that our region-based gene identification method can reliably identify FpvA receptors.

Next, we explored the diversity among FpvA receptors in greater detail by focusing on the 1534 strains with proximate receptors located within 20 kb of the pyoverdine Pep genes (*Figure 6d*). We

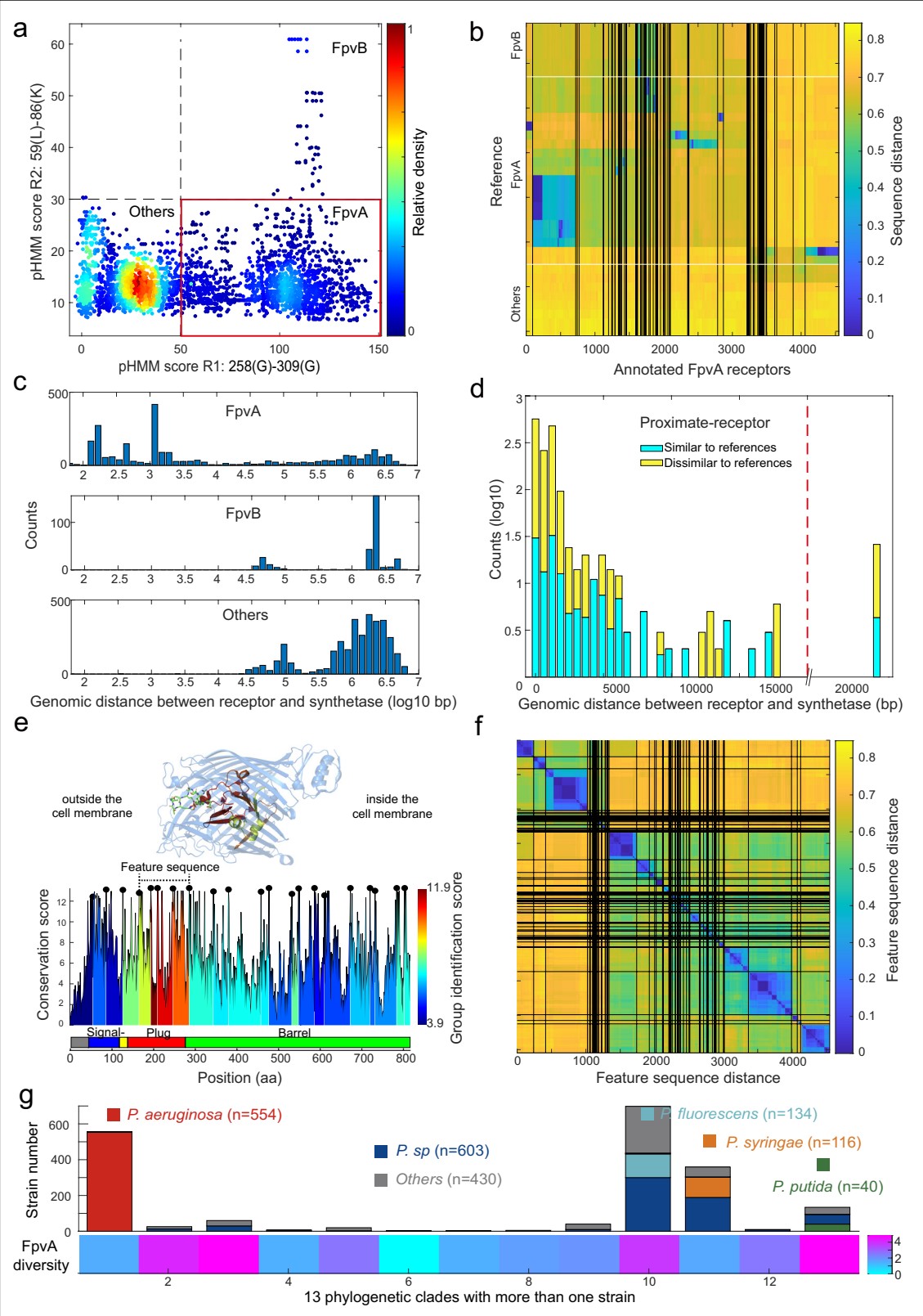

**Figure 6.** Application of the receptor annotation pipeline to the full database. (**a**) Applying the receptor annotation pipeline to the genomes of the 1928 non-redundant *Pseudomonas* strains yields 14301 Fpv-like receptors, which segregate into 4547 FpvA receptors (red box), 615 FpvB receptors, and 9139 other receptors, based on the profile hidden Markov model (pHMM) score thresholds for regions R1 and R2. The heatmap indicates receptor density. (**b**) Sequence distance matrix between the 35 reference sequences (y-axis) and the 4547 annotated FpvA sequences in the full database (x-axis). Database

*Figure 6 continued on next page*

*Figure 6 continued*

sequences were ordered by hierarchically clustering and segregated into 114 groups. 2254 of the annotated FpvA sequences have low sequence identity <60% compared to the reference receptors, pointing at many novel subtypes of FpvA receptors. (**c**) Genomic distance (in base pairs) between each Fpv-like receptor sequence and its pyoverdine peptide synthetase gene (Pep) for annotated FpvA receptors (upper panel), FpvB receptors (middle panel), and other receptors (lower panel). (**d**) Distribution of the genomic distance between each FpvA receptor and its nearest pyoverdine peptide synthetase depending on whether the annotated FpvA receptor has high sequency similarity (blue, ≥50%) or low sequence similarity (yellow, <50%) with at least 1 of the 21 reference FpvAs. (**e**) FpvA region-based conservation scores from a multi-alignment of all the annotated FpvA receptors that are proximate (<20 kbp) to the pyoverdine synthetase cluster mapped to the FpvA sequence of strain *P. aeruginosa* PAO1. All residues within the top 10% of the conservation score are denoted with black dots. For each region flanked by two black dots, we calculated the group identification score (heatmap, lower panel), representing the ability of the region to distinguish between different groups of FpvA receptors. Four regions in the plug domains had a particularly high group identification score (called the feature sequence). They are mapped to the crystal structure of FpvA from PAO1 conjugated with pyoverdine (PDB 2IAH, upper panel). All four regions surround the pyoverdine transmission channel and are shown in the respective heatmap color. (**f**) Heatmap showing the hierarchically clustered distances between the 4547 annotated FpvA receptors based on the feature sequence (comprising the four groups with the highest identification scores). The analysis identifies 94 receptor groups with a 70% identity threshold. (**g**) The diversity of FpvA receptors along the 13 phylogeny clades containing more than 1 strain. Receptor diversity was calculated by the Shannon entropy, similar to the alpha-diversity in microbial community.

The online version of this article includes the following figure supplement(s) for figure 6:

**Figure supplement 1.** Comparison of clustering and grouping effects of all proximate-receptors.

**Figure supplement 2.** Applying the receptor annotation pipeline to >30,000 complete bacterial genomes in NCBI based on the profile hidden Markov model (pHMM) score thresholds for regions R1 and R2.

**Figure supplement 3.** The distribution of 31,936 FpvAs spans various taxonomic categories.

utilized these high-confidence FpvAs for sequence feature extraction. When considering the whole gene sequences, these receptors segregated into 44 groups according to single-linkage clustering with an identity threshold of 60% (*Figure 6—figure supplement 1a*). To investigate which sequence regions were the most informative for reliable clustering, we used a similar approach as with FpvAs detection by quantifying the 'group identification score' for variable regions flanked by highly conserved sites. The higher the score, the stronger a region's capacity to discriminate between FpvA groups. We found that the four regions with the top discrimination capacities all located near the Plug domain surrounding the pyoverdine transmission channel (*Figure 6e*). The plug domain is known to undergo conformational changes and is involved in pyoverdine selectivity and import (*Schalk et al., 2012*; *Greenwald et al., 2009*), suggesting that the four high-score regions are responsible for pyoverdine specificity.

Based on the above insights, we concatenated the four high-score regions (from 168 Pro to 295 Ala in PAO1) into a single 'feature sequence'. The feature sequence could characterize 98% of the distance matrix compared to the whole sequence (1534 FpvAs, *r*=0.98, *Figure 6—figure supplement 1*) and substantially reduced within-group distance. We applied the concatenated feature sequence approach to all the 4547 annotated FpvAs to calculate the sequence distance matrix. Single-linkage clustering with an identity threshold of 70% revealed a total of 94 groups, out of which 43 groups contained more than 10 members (*Figure 6f*). The diversity of receptors is hence much larger than currently anticipated as only three groups of FpvAs have previously been reported. Finally, we calculated the diversity of receptor FpvAs for each of the 13 phylogenetic clades with more than one strain by the Shannon entropy, which is similar to the alpha-diversity in microbial community (*Figure 6g*). We noticed large differences in FpvA diversity across the clades and species: clades with *P. aeruginosa* and *Pseudomonas syringae* species had lower FpvAs diversity (1.55 and 1.60) than clades containing *Pseudomonas putida* and *Pseudomonas fluorescens* species (4.82 and 3.77). Taken together, the region-based identification method developed in our study can reliably mine pyoverdine receptors from genome data, revealing an undiscovered FpvA diversity that is unequally distributed across the different phylogenetic clades of pseudomonads.

We further extended the search for FpvA receptors, using our pipeline in *Figure 5h*, to the >30,000 complete bacterial genomes in NCBI. We found 31,936 receptors to pass the R1 and R2 thresholds (*Figure 6—figure supplement 2*), and surprisingly found that FpvA receptors seem to be widely dispersed across 12,944 strains spanning 13 phyla, 22 classes, 54 orders, 139 families, 468 genera, and 2598 species (*Figure 6—figure supplement 3* and *Supplementary file 11*). While *Pseudomonas* remains the genus with the highest distribution of FpvAs, other genera with notable distributions

comprise *Escherichia*, *Salmonella*, *Enterobacter*, *Acinetobacter*, *Xanthomonas*, *Bordetella*, and *Achromobacter*. While confirmatory experimental work is still pending, these analyses suggest that FpvA receptors for pyoverdine uptake may be widely distributed throughout the bacterial kingdom.

## Discussion

The rapid expansion of sequencing data offers exciting opportunities for microbiology (*Almeida et al., 2019*; *Schloss and Handelsman, 2005*; *Handelsman, 2004*). One key challenge of current research in the field is to infer biological functions of microbial communities from genome sequence data (*Tsilimigras and Fodor, 2016*; *Weiss et al., 2016*; *Faust and Raes, 2012*). While this endeavor is increasingly successful for biological functions involving the primary metabolism and the associated complex metabolic flux, reconstructing aspects of the secondary metabolism is much more challenging. The main issue is that neither the function of a secondary metabolite enzyme nor the resulting metabolite can be precisely predicted from gene sequence data. In our study, we tackled this challenge and developed a bioinformatic pipeline to reconstruct the complete secondary metabolism pathway of pyoverdines, a class of iron-scavenging siderophores produced by *Pseudomonas* spp. These secondary metabolites are synthetized by a series of non-ribosomal peptide synthetases and require a specific receptor (FpvA) for uptake. We combined knowledge-guided learning with phylogeny-based methods to predict with high accuracy: (i) the full pyoverdine assembly line, (ii) the substrate specificity for each enzyme within the assembly lines, (iii) the complete chemical structure of pyoverdines, and (iv) the FpvA receptors from genome sequences. After validation, we tested our pipeline with sequence data from 1664 phylogenetically distinct *Pseudomonas* strains and were able to determine 18,292 enzymatic A domains involved in pyoverdine biosynthesis, to reliably predict 97.8% of their substrates, and to identify 188 different pyoverdine molecule structures and 4547 FpvA receptor variants belonging to 94 distinct groups. The uncovered diversity is stunning and goes far beyond currently known levels of variation (73 pyoverdines and 3 FpvA groups).

We show that knowledge-guided learning is an extremely powerful tool to predict enzyme, metabolite, and receptor properties. The establishment of our entire pipeline is based on only 101 previously known enzymatic A domains (from 13 known pyoverdine assembly lines) and 21 FpvA receptor sequences. Even with this limited amount of information, we were able to predict the substrates of almost all the 18,292 enzymatic A domains and to identify 4547 FpvA receptors from the sequence data. A key insight from our knowledge-guided learning is that comparisons based on the full gene sequences (e.g. for pyoverdine synthetase or receptor) are typically non-informative and unsuitable for obtaining functional information. This is because overall diversity does not stand for functional diversity, meaning that A domains recognizing the same substrate can diverge substantially in their full sequences. The same holds true for receptor sequences: whole-sequence alignments can neither accurately identify FpvA receptors nor reliably separate them into functional groups. Instead, it is imperative to extract informative feature sequences that are defined as sequence stretches within a gene whose diversity is tightly linked to variation in its functioning. We successfully extracted and applied feature sequence comparisons for both A domain substrate prediction and FpvA identification. It is important to note that a knowledge-guided pipeline does not have to be perfect right from the start. For example, our pipeline for pyoverdine structure prediction returned unknowns for several amino acid positions within the PEP. Our experimental verifications then revealed new substrates such as valine and citrulline. This information can then be used to refine our prediction algorithm in a feedback loop.

Another main advantage of our bioinformatic pipeline is that it can be applied to draft genomes. This reflects a major improvement compared to existing annotation tools such as antiSMASH (*Blin et al., 2019*), which typically has difficulties in recognizing NRPS structures in fragmented genome assemblies. However, draft genomes are the most common data source in microbiology. While our pipeline shows high performance, we need to acknowledge that we still lose many genomes (6087 out of 9599, 63.4%). The reason for the loss is that the pyoverdine biosynthesis machinery is large, which increases the probability that it is positioned at the end of a contig. We decided to exclude those cases because the annotated biosynthesis machinery might be truncated and thus incomplete. Thus, the high loss rate of draft genomes is rather due to limitations in sequence quality (too many short contigs) and not due to a limitation of our bioinformatic pipeline. We believe that this limitation

may disappear in the future as long-read sequencing technologies are quickly becoming cheaper and more reliable.

We further show that knowledge-guided learning combined with a phylogeny-focused approach is a powerful tool for predicting the substrate specificity of A domains of synthetases. It outperforms currently known bioinformatics prediction tools of NRPS substrates such as antiSMASH (*Blin et al., 2019*). Most current algorithms (*Khayatt et al., 2013*; *Minowa et al., 2007*; *Prieto et al., 2012*; *Röttig et al., 2011*; *Zierep et al., 2020*) perform poorly when applied to pyoverdines, particularly when encountering non-proteogenic amino acids. The high accuracy of our algorithm can largely be attributed to our reference set, composing only 13 pyoverdines from *Pseudomonas* spp., yet capturing most of the substrate diversity. Similarly accurate predictions based on a handful of known substrates among closely related species were observed in several fungal NRPS systems (*Fan et al., 2022*). It is worth noting that when the algorithm output is 'unknown', it signifies the presence of uncharacterized A domains that are not yet incorporated into the reference dataset. This should prompt researchers to pay attention to these A domains, and like in our case, subject them to further experimental investigation. This approach helped us discover new substrates (valine, citrulline) and link them to the corresponding A domains. The novel substrates identified through our structural assessment and mass spectrometry experiment can subsequently be used to enhance the precision of our phylogeny-centered substrate prediction technique in the future, creating a progressive feedback loop of expanding knowledge. Taken together, supervised learning based on a few known compounds produced by species from the same genus can outperform generalized prediction algorithms trained on many products from a diverse set of microbes for NRPS substrate predictions.

Our results show that both pyoverdine and receptor diversity has been vastly underestimated. While considerable pyoverdine diversity (*n*=73) has already been captured in previous studies, here we predicted 151 putatively new variants. On the receptor side, the uncovered novel diversity is more dramatic. One reason for this is that research on receptors has mainly focused on the pathogen *P. aeruginosa* (*Cobessi et al., 2005*; *Diggle and Whiteley, 2020*; *Meyer et al., 1997*; *Bodilis et al., 2009*; *Smith et al., 2005*). For this species, three different pyoverdine types were described (*Meyer et al., 1997*) together with three structurally different FpvA receptor types that each recognize one of the pyoverdine types (*Bodilis et al., 2009*). While our study confirmed that *P. aeruginosa* strains (*n*=554) indeed have only 3 pyoverdine-receptor types, we further discovered 91 new FpvA groups among environmental *Pseudomonas* spp. Our findings raise the question why there are so many different pyoverdine and receptor variants. One potential explanation is that the benefit of specific siderophores could be context-dependent and locally adapted to the multitude of different environmental conditions pseudomonads are exposed to. For example, experimental work has revealed that pyoverdines can be cooperatively shared among strains with matching receptors (*Kramer et al., 2020*; *Gu et al., 2020*), or conversely, pyoverdines can serve as competitive agents by locking away iron from species that have non-matching receptors (*Figueiredo et al., 2022*). Given that bioavailable iron is limited in most natural and host-associated habitats (*Andrews et al., 2003*; *Boyd and Ellwood, 2010*; *Emerson et al., 2012*), the unraveled functional diversity is likely a direct evolutionary consequence of the struggle and competition of microbes for iron. While experimental work is often restricted to a low number of strains, we propose that our bioinformatic pipeline can be used to predict pyoverdine-mediated interaction networks across thousands of strains and across different habitats. We have addressed this point in a complementary study (*Gu et al., 2024*).

We are confident that our pipeline can be expanded to study iron competition in multi-species communities and perhaps also in plant-microbe ecosystems, as siderophores exist ubiquitously and are shared among microbes and certain hosts (*Ruolin, 2023*). To move further, a key question is whether our knowledge-guided approach can be applied to other microbes and important secondary metabolites, such as antibiotics, toxins, biosurfactants, and pigments? Our preliminary investigation, focused on the curated dataset in *Burkholderiales* as a test case, provides a highly promising response to this query. The outcomes from this evaluation predominantly showcased the method's potential for extension to diverse microbial genera and metabolites, given sufficiently accurate dataset. Meanwhile, we observed a notable decline in accuracy (from 94% to 83%) when compared to pyoverdine. This indicates that our algorithm possibly needs refinement, particularly concerning specific secondary metabolites or microbial genera. As soon as sufficient case-by-case knowledge on a specific system is available, the annotation strategies together with the feature sequence extraction

and the phylogeny-focused approach developed in our paper can be applied. It is worth noting that our current algorithm does not predict tailoring reactions. While tailoring reactions are crucial for predicting the final NRPS product structure, this challenge is not addressed by any existing pipelines and remains an area for future research. In our study, it is important to note that the specificity of pyoverdines is mainly determined by the backbone composition, with tailoring reactions playing a relatively minor role (*Ringel and Brüser, 2018*). For most of the secondary metabolites listed above, there are no receptors as the compounds have purely extracellular functions, which substantially simplifies the development of bioinformatic pipelines. In the long run, it will certainly be possible to automate the steps implemented in our workflow so that the algorithms can be applied to a large set of secondary metabolites when fed with an appropriate training set.

## Methods

### Construction of phylogeny tree

The phylogenetic tree depicted in *Figure 1e* was constructed utilizing the PhyloPhlAn3 pipeline (*Asnicar et al., 2020*). PhyloPhlAn is a comprehensive pipeline that encompasses the identification of phylogenetic markers, MSA, and the inference of phylogenetic trees. In this analysis, we employed 400 universal genes defined by PhyloPhlAn as our selected phylogeny markers. Subsequently, the taxonomic cladogram was generated using the iTOL web tool (http://huttenhower.sph.harvard.edu/galaxy/).

### Application of our pipeline to identify errors in the substrate information associated with the A domain of *Pseudomonas* in the MIBiG database

We downloaded all A domain and substrate data for all NRPS secondary metabolism instances for Pseudomonas within MIBiG 3.0 (*Terlouw et al., 2023*; *Supplementary file 4*). Following our pipeline, we initiated the process by utilizing NRPSMotifFinder to extract all Amotif4-5 sequences of the A domains. Subsequently, we calculated the Jukes-Cantor sequence distance between Amotif4-5 pairs. Finally, employing Ward linkage clustering, we conducted a cluster analysis on all Amotif4-5 sequences, resulting in the generation of *Figure 3—figure supplement 1*. We observed that within the same cluster, substrates often exhibit some subtle impurities. We speculate that such clusters might contain some errors in substrate information. Consequently, we focused on these potential errors, manually searched for the respective original literature, and verified that the A domain and its corresponding substrate information entered in MIBiG indeed contain errors (*Supplementary file 5*).

### Application of our pipeline to predict the structural composition of diverse NRPS metabolites across various genera of *Burkholderiales*

To evaluate the extendibility of our bioinformatic pipeline to various metabolites and microbial genera, we utilized a previously curated dataset of A domain sequences with experimentally confirmed substrates for *Burkholderiales* (*Chen et al., 2023*; *Supplementary file 7*). This manually curated dataset, encompassing 7 genera, 34 secondary metabolite products, 203 A domains, and 25 substrates within *Burkholderiales*, served as a valuable resource and as a test set to validate the extendibility of our bioinformatic approach. Simultaneously, we compiled a training set by aggregating A domain and substrate data from all NRPS secondary metabolism instances within *Burkholderiales*, utilizing experimentally validated biosynthetic gene cluster data from MIBiG 3.0 (*Terlouw et al., 2023*; *Supplementary file 8*). This initial training set covered 6 genera, 19 secondary products, 99 A domains, and 21 substrates. To ensure the inclusivity of all substrates from the test set, we randomly selected an A domain from each substrate in the manually curated dataset, incorporating it into our training set. Consequently, the finalized training set comprised 124 A domains (*Supplementary file 12*), while the test set encompassed 178 A domains (*Supplementary file 9*). Then, we concurrently employed our pipeline and antiSMASH (*Blin et al., 2023*) to predict the corresponding substrate based on the sequence data of the A domain in the test set. Subsequently, we compared these predictions with the actual substrate information, calculating the accuracy rate for the two algorithms.

### Application of our pipeline to annotate FpvA receptors across the bacterial domain

To explore the distribution of FpvA receptors across the bacterial domain, we acquired all complete bacterial genomes (33,207) from NCBI (https://www.ncbi.nlm.nih.gov). Employing the pipeline

depicted in *Figure 5h* to examine the CDS of these genomes, we identified a total of 357,790 TonB-dependent receptors, of which 31,936 and 623 matched the criteria for being considered as FpvA and FpvB receptors, respectively. Subsequently, leveraging the taxonomy information from NCBI, we conducted a comparative analysis of the distribution differences between FpvA and FpvB among various species. The findings suggest that FpvA exhibit a widespread distribution across different species, whereas FpvB was exclusive to the genus *Pseudomonas*, with a maximum occurrence of one copy per genome.

## Acknowledgements

We thank Richard Allen and Simon Maréchal for the handling of the genome sequencing of the 20 *Pseudomonas* strains used for the structure validation experiment. We further thank Vera Vollenweider for sample preparation for the experiment elucidating pyoverdine structures. This work was supported by the National Key Research and Development Program of China (No. 2021YFF1200500, 2021YFA0910700), National Natural Science Foundation of China (No. 42107140, No. 32071255, No.41922053, No. T2321001), National Postdoctoral Program for Innovative Talents (No. BX2021012). RK is supported by a grant from the Swiss National Science Foundation no. 310030_212266. VPF is supported jointly by a grant from UKRI, Defra, and the Scottish Government, under the Strategic Priorities Fund Plant Bacterial Diseases program (BB/T010606/1), Research Council of Finland, and The Finnish Research Impact Foundation.

## Additional information

### Funding

| Funder | Grant reference number | Author |
|---|---|---|
| National Key Research and Development Program of China | No. 2021YFF1200500 | Zhiyuan Li |
| National Natural Science Foundation of China | No. 42107140 | Shaohua Gu |
| National Natural Science Foundation of China | No.41922053 | Zhong Wei |
| National Natural Science Foundation of China | No. 32071255 | Zhiyuan Li |
| National Natural Science Foundation of China | No. T2321001 | Zhiyuan Li |
| National Postdoctoral Program for Innovative Talents | No. BX2021012 | Shaohua Gu |
| Swiss National Science Foundation | no. 310030_212266 | Rolf Kümmerli |

The funders had no role in study design, data collection and interpretation, or the decision to submit the work for publication.

### Author contributions

Shaohua Gu, Conceptualization, Data curation, Formal analysis, Funding acquisition, Validation, Investigation, Visualization, Methodology, Writing – original draft, Project administration, Writing – review and editing; Yuanzhe Shao, Data curation, Software, Formal analysis, Investigation, Visualization, Methodology; Karoline Rehm, Laurent Bigler, Resources, Data curation, Validation; Di Zhang, Software, Formal analysis, Visualization, Methodology; Ruolin He, Software, Methodology; Ruichen Xu, Investigation; Jiqi Shao, Investigation, Methodology; Alexandre Jousset, Ville-Petri Friman, Writing – review and editing; Xiaoying Bian, Resources, Investigation; Zhong Wei, Conceptualization, Resources, Supervision, Funding acquisition, Validation, Methodology, Project administration, Writing

– review and editing; Rolf Kümmerli, Conceptualization, Resources, Supervision, Funding acquisition, Validation, Methodology, Writing – original draft, Project administration, Writing – review and editing; Zhiyuan Li, Conceptualization, Resources, Data curation, Software, Formal analysis, Supervision, Funding acquisition, Investigation, Methodology, Writing – original draft, Project administration, Writing – review and editing

### Author ORCIDs
Shaohua Gu http://orcid.org/0000-0003-3071-5747
Yuanzhe Shao https://orcid.org/0009-0003-6739-4198
Zhong Wei https://orcid.org/0000-0002-7967-4897
Rolf Kümmerli https://orcid.org/0000-0003-4084-6679
Zhiyuan Li https://orcid.org/0000-0001-6662-2636

Reviewer #1 (Public Review): https://doi.org/10.7554/eLife.96719.3.sa1
Reviewer #2 (Public Review): https://doi.org/10.7554/eLife.96719.3.sa2
Author response https://doi.org/10.7554/eLife.96719.3.sa3

## Additional files

### Supplementary files
• Supplementary file 1. The information of 73 pyoverdine structures have been reported.

• Supplementary file 2. 13 *Pseudomonas* strains for which both pyoverdine structures and completely sequenced synthetase genes were available.

• Supplementary file 3. The sequences and producing strains of 101A domains that are linked to 13 experimentally confirmed amino acid substrates.

• Supplementary file 4. All A domain and substrate data for all non-ribosomal peptide synthetase (NRPS) secondary metabolism instances for *Pseudonomas* within MIBiG 3.0.

• Supplementary file 5. The A domains with incorrect corresponding substrate information in MIBiG 3.0.

• Supplementary file 6. The results of comprehensive comparative analysis of our method with six different commonly used methods, including NP.searcher, PRISM4, AdenPredictor, SeMPI2, SANDPUMA, antiSMASH5.

• Supplementary file 7. A previously curated dataset of A domain sequences with experimentally confirmed substrates for *Burkholderiales*.

• Supplementary file 8. All A domain and substrate data for all non-ribosomal peptide synthetase (NRPS) secondary metabolism instances for *Burkholderiales* within MIBiG 3.0.

• Supplementary file 9. The test set encompassed 178A domain sequences with experimentally confirmed substrates for *Burkholderiales*.

• Supplementary file 10. 21 FpvA, 6 FpvB, and 8 TonB-dependent siderophore receptor sequences often found in *Pseudomonas* genomes, encoding receptors for the uptake of heterologous siderophores.

• Supplementary file 11. Species information of the genome that can be annotated to the FpvA gene.

• Supplementary file 12. The finalized training set comprised 124A domain sequences with experimentally confirmed substrates for *Burkholderiales*.

• Supplementary file 13. MS/MS profiles of 14 pyoverdines.

• MDAR checklist

### Data availability
The source code and parameters used are available in the supplementary material. The sequences of the 20 Pseudomonas strains are available in ENA under accession number PRJEB76792. The raw data and code can be accessed at the Dryad repository (https://doi.org/10.5061/dryad.jdfn2z3kw) and on GitHub (https://github.com/zhiyuanLab/Pyoverdine-genome-mining copy archived at *Gu, 2024*).

The following datasets were generated:

| Author(s) | Year | Dataset title | Dataset URL | Database and Identifier |
|---|---|---|---|---|
| Maréchal S, Kümmerli R | 2024 | From sequence to molecules: Feature sequence-based genome mining uncovers the hidden diversity of bacterial siderophore pathways | https://www.ebi.ac.uk/ena/browser/view/PRJEB76792 | European Nucleotide Archive, PRJEB76792 |
| Gu S, Li Z | 2024 | Data from: Feature sequence-based genome mining uncovers the hidden diversity of bacterial siderophore pathways | https://doi.org/10.5061/dryad.jdfn2z3kw | Dryad Digital Repository, 10.5061/dryad.jdfn2z3kw |

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
