## [Editor Report · eLife assessment]

This **important** study presents a novel pipeline for the large-scale genomic prediction of members of the non-ribosomal peptide group of pyoverdines based on a dataset from nearly 2000 Pseudomonas genomes. The advance presented in this study is based on **convincing** evidence. This study of bacterial siderophores has broad theoretical and practical implications beyond a singular subfield.

---

## [Referee Report · Reviewer #1 (Public Review)]

The manuscript introduces a bioinformatic pipeline designed to enhance the structure prediction of pyoverdines, revealing an extensive and previously overlooked diversity in siderophores and receptors. Utilizing a combination of feature sequence and phylogenetic approaches, the method aims to address the challenging task of predicting structures based on dispersed gene clusters, particularly relevant for pyoverdines.

Predicting structures based on gene clusters is still challenging, especially pyoverdines as the gene clusters are often spread to different locations in the genome. The revised manuscript has much improved in clarity and reproducibility. I believe that the method is not yet applicable to all NRPS in general and that there is a clear scalability issue when talking about Big Data. However, the method is highly useful for specific NRPS families such as the pyoverdines, so the manuscript presents a useful bioinformatic pipeline for pyoverdine structure prediction, showcasing a commendable exploration of siderophore diversity.

---

## [Referee Report · Reviewer #2 (Public Review)]

Pyoverdines, siderophores produced by many Pseudomonads, are one of the most diverse groups of specialized metabolites and frequently used as model systems. Thousands of Pseudomonas genomes are available, but large scale analyses of pyoverdines are hampered by the biosynthetic gene clusters (BGCs) being spread across multiple genomic loci and existing tools' inability to accurately predict amino acid substrates of the biosynthetic adenylation (A) domains. The authors present a bioinformatics pipeline that identifies pyoverdine BGCs and predicts the A domain substrates with high accuracy. They tackled a second challenging problem by developing an algorithm to differentiate between outer membrane receptor selectivity for pyoverdines versus other siderophores and substrates. The authors applied their dataset to thousands of Pseudomonas strains, producing the first comprehensive overview of pyoverdines and their receptors and predicting many new structural variants.

The A domain substrate prediction is impressive, including the correction of entries in the MIBiG database. Their high accuracy came from a relatively small training dataset of A domains from 13 pyoverdine BGCs. The authors acknowledge that this small dataset does not include all substrates, and correctly point out that new sequence/structure pairs can be added to the training set to refine the prediction algorithm. The workflow unfortunately cannot differentiate between different variants of Asp and OHOrn. To validate their predictions, they elucidated structures of several new pyoverdines, and their predictions performed well. The authors tested their workflow on Burkholderiales A domains and had good results, suggesting it can be used on other taxa. Skimming through the source code and data, the algorithm itself appears to be sound and a clear improvement over existing tools for pyoverdine BGC annotation.

Predicting outer membrane receptor specificity is likewise a challenging problem and the authors have made a promising achievement by finding specific gene regions that differentiate the pyoverdine receptor FpvA from FpvB and other receptor families. Their predictions were not tested experimentally, but the finding that only predicted FpvA receptors were proximate to the biosynthesis genes lends credence to the predictive power of the workflow. The authors find predicted pyoverdine receptors across an impressive 468 genera, an exciting finding for expanding the role of pyoverdines as public goods beyond Pseudomonas. However, whether or not these receptors can actually recognize pyoverdines (and if so, which structures!) remains to be investigated.

In all, the authors have assembled a rich dataset that will enable large scale comparative genomic analyses. This dataset could be used by a variety of researchers, including those studying natural product evolution, public good eco/evo dynamics, and NRPS engineering.

---

## [Author Response]

The following is the authors’ response to the original reviews.

**eLife assessment**
This important study presents a novel pipeline for the large-scale genomic prediction of members of the non-ribosomal peptide group of pyoverdines based on a dataset from nearly 2000 Pseudomonas genomes. The advance presented in this study is largely based on solid evidence, although some main claims are only incompletely supported. This study on bacterial siderophores has broad theoretical and practical implications beyond a singular subfield.

Thank you for the supportive and encouraging words. We appreciate the editor’s and reviewers’ careful and professional assessment of this manuscript. The reviewers’ scrutiny has helped us to improve the presentation and discussion of our work. We have now carefully revised the manuscript following their instructive suggestions and comments. Please find below our detailed responses (marked in blue) to each of the comments.

**Public Reviews:**

**Reviewer #1 (Public Review):**
The manuscript introduces a bioinformatic pipeline designed to enhance the structure prediction of pyoverdines, revealing an extensive and previously overlooked diversity in siderophores and receptors. Utilizing a combination of feature sequence and phylogenetic approaches, the method aims to address the challenging task of predicting structures based on dispersed gene clusters, particularly relevant for pyoverdines.Predicting structures based on gene clusters is still challenging, especially pyoverdines as the gene clusters are often spread to different locations in the genome. An improved method would indeed be highly useful, and the diversity of pyoverdine gene clusters and receptors identified is impressive.However, so far the method basically aligns the structural genes and domains involved in pyoverdine biosynthesis and then predicts A domain specificity to predict the encoded compounds. Both methods are not particularly new as they are included in other tools such as PRISM (10.1093/nar/gkx320) or Sandpuma (https://doi.org/10.1093/bioinformatics/btx400) among others. The study claims superiority in A domain prediction compared to existing tools, yet the support is currently limited, relying on a comparison solely with AntiSMASH. A more extensive and systematic comparison with other tools is needed.

Thanks for pointing this out. In the revised manuscript, we have included a comprehensive comparative analysis, in which we compared our pipeline to six different commonly used methods, including NP.searcher, PRISM4, AdenPredictor, SeMPI2, SANDPUMA, antiSMASH5 (see Supplementary_table 6 for details, and lines 281-286). These approaches either consist of a single specific algorithm or integrate several methods. Our approach performs best (see table below), demonstrating a clear improvement over previous tool. The improvements are due to several methodological differences inherent to our approach. Additionally, while exploring existing prediction tools, we found that some had not been maintained for years. For instance, we were unable to access NRPSsp (http://www.nrpssp.com) and NRPSpredictor2 (http://nrps.informatik.uni-tuebingen.de/). Below, we briefly explain these differences, particularly in relation to PRISM and SANDPUMA, as highlighted by the reviewer.

**Author response table 1. sa3table1:** 

	Description	Accuracy
PHYfocus (Our method)	phylogeny focus	94%
NP.searcher	specificity code	63%
PRISM4	phmm	79%
AdenPredictor	Mix	38%
SeMPI2	randomforest	58%
SANDPUMA	Mix	46%
antiSMASH5	Mix	59%

PRISM annotates biosynthetic gene clusters (BGC) and reconstructs the linear structures of NRPS synthetases, with this function depending on proper annotations of open reading frames. This pipeline can have difficulties in assembling the linear structure into a final product. In our approach, we found that the annotations of NRPS gene are frequently truncated because of sequencing errors and annotation issues. Our method fixes this problem through rescanning all possible reading frames of the BGC to rebuild complete pyoverdine synthetase genes.

Sandpum and our approach are based on similar ideas (using the prediCAT algorithm) to predict A domain substrates, namely by using the closest reference A domain annotated. However, our method uses a self-adaptive feature extraction step to reduce the co-founding influence of phylogeny. This small adjustment significantly improves the performance of our approach and even works well for small training sets (101 experimentally validated A domains with our approach as opposed to 494 A domains used by Sandpuma from MIBiG).

Additionally, in contradiction to the authors' claims, the method's applicability seems constrained to well-known and widely distributed gene clusters. The absence of predictions for new amino acids raises concerns about its generalizability to NRPS beyond the studied cases.

We thank the reviewers for this comment. We acknowledge that our method cannot directly predict new amino acids. Nevertheless, for several reasons we believe that our approach is not constrained and can be widely applied in the future.

First, our method can identify A domains that select new unknown amino acid substrates. In fact, three of the four unresolved cases in our experimental verification analysis (Fig. 3d) represent new amino acids. Obviously, experimental verification is required to characterize the unknown substrate. Once verified, the new A domains and their substrates can expand the reference dataset, allowing targeted improvement of our phylogeny-focused prediction technique. We now discuss this aspect in lines 634-645.

Second, despite that the overall substrate diversity in NRPS is high across the microbial kingdom, our analysis suggests that the number of amino acids used for a specific group of secondary metabolites quickly reaches a saturation point. The discovery rate of new amino acids was 1.7% for our experimental *Pseudomonas* data set (Fig. 3d). The discovery rate of new amino acids was even 0.0 % for the *Burkholderiales* data set. This suggests that as the database expands, the discovery rate of novel amino acid substrates is expected to drop rapidly.

Third, we acknowledge that the inability to predict the substrates of unknown domains is a common limitation among all knowledge-guided learning algorithms, including ours. However, we have made significant improvements in prediction accuracy. As the database grows, we expect the rate of unknown substrates to decrease, and the prediction accuracy to increase.

The manuscript lacks clarity on how the alignment of structural genes operates when dealing with multiple NRPS gene clusters on different genome contigs. How would the alignment of each BGC work?

We thank the reviewers for this comment. The pyoverdine molecules consist of a conserved fluorescent chromophore (Flu) and a peptide chain (Pep), both synthesized by NRPS enzymes. In most instances (over 90%), Flu and Pep are produced by two separate biosynthetic gene clusters (BGCs). In these cases, we merge the two BGCs by positioning Flu at the head and Pep at the tail. For the remaining less than 10%, there are two scenarios: 1. Flu and Pep are located on the same BGC, which eliminates any issues with BGC alignment. 2. In very rare cases, Flu and Pep are synthesized by three BGCs. Here, Flu is still synthesized by one BGC at the head, while Pep is produced by two BGCs. We put the BGC containing the Thioesterase (TE) domain as the tail and the BGC not containing the TE domain in the middle.

(see lines 165-169).

Another critical concern is that a main challenge in NRPS structure prediction is not the backbone prediction but rather the prediction of tailoring reactions, which is not addressed in the manuscript at all, and this limitation extensively restricts the applicability of the method.

While we thank the reviewer for this comment, we only partly agree with it. Peptide backbone predictions are still a significant challenge. This challenge is clearly visible in our new analysis comparing prediction accuracies of different pipelines, such as antiSMASH5, PRISM4, AdenPredictor, SeMPI2, NP.searcher, Sandpuma. Unresolved and wrong substrate predictions are still common, highlighting the importance of our contribution in developing a new approach with improved high accuracy.

However, we agree with the reviewer that our current algorithm does not predict tailoring reactions (now discussed on lines 680-685). Although tailoring reactions are important for predicting the final NRPS product structure, none of the other existing pipelines address this issue either, and it remains a challenge for future work. For our study, it is important to note that the specificity of pyoverdines is primarily determined by the backbone composition, whereas tailoring reactions seem to play a minor role.

The manuscript presents a potentially highly useful bioinformatic pipeline for pyoverdine structure prediction, showcasing a commendable exploration of siderophore diversity. However, some of the claims made remain unsubstantiated. Overall, while the study holds promise, further validation and refinement are required to fulfill its potential impact on the field of bioinformatic structure prediction.

Thank you for the supportive and encouraging words. We deeply appreciate your constructive comments and suggestions.

**Reviewer #2 (Public Review):**
Pyoverdines, siderophores produced by many Pseudomonads, are one of the most diverse groups of specialized metabolites and are frequently used as model systems. Thousands of Pseudomonas genomes are available, but large-scale analyses of pyoverdines are hampered by the biosynthetic gene clusters (BGCs) being spread across multiple genomic loci and existing tools' inability to accurately predict amino acid substrates of the biosynthetic adenylation (A) domains. The authors present a bioinformatics pipeline that identifies pyoverdine BGCs and predicts the A domain substrates with high accuracy. They tackled a second challenging problem by developing an algorithm to differentiate between outer membrane receptor selectivity for pyoverdines versus other siderophores and substrates. The authors applied their dataset to thousands of Pseudomonas strains, producing the first comprehensive overview of pyoverdines and their receptors and predicting many new structural variants.The A domain substrate prediction is impressive, including the correction of entries in the MIBiG database. Their high accuracy came from a relatively small training dataset of A domains from 13 pyoverdine BGCs. The authors acknowledge that this small dataset does not include all substrates, and correctly point out that new sequence/structure pairs can be added to the training set to refine the prediction algorithm.The authors could have been more comprehensive in finding their training set data. For instance, the authors claim that histidine "had not been previously documented in pyoverdines", but the sequenced strain P. entomophila L48, incorporates His (10.1007/s10534-009-9247-y).

Thank you for highlighting this issue. We agree that stating histidine has not been reported before in pyoverdine was incorrect. We have reviewed the full text and made the necessary corrections.

The primary reason for excluding the sequenced strains *P. syringae* 1448a (10.1186/14712180-11-218) and *P. entomophila* L48 (10.1007/s10534-009-9247-y) from the training set is that the pyoverdine structures of these strains were not determined solely through experimental methods. In these works, the pyoverdine structures were predicted based on the synthetic gene sequence using bioinformatical analysis, followed by structural analysis experiments based on this predicted structure. We found that pre-prediction probably has introduced biases into downstream analyses. Specifically, in the case of *Pseudomonas entomophila* L48, we discovered inaccuracies in the annotation of certain domains (see figures below). For example, the third A domain of the peptide chain in *P. entomophila* L48 pyoverdine was initially annotated with Dab specificity. However, upon closer examination, it appears to differ significantly from other Dab references (top) or Dab from our experimentally validated (right) domains (left panel in the figure below). By analyzing the interface (I) domain (10.1073/pnas.1903161116) in its predicted site, we suggested that it should actually recognize OHHis. The OHAsp domain of *P. entomophila* L48 reported in the paper is actually close in sequence similarity to the OHAsp domain (left panel in the figure below), while the Ala domain reported is more similar to the Ser domain (right panel in the figure below). For these reasons, we did not include this supervised pyoverdine structure analysis strain in the training set data.

The workflow cannot differentiate between different variants of Asp and OHOrn, and it's not clear if this is a limitation of the workflow, the training data, or both.

Thanks for pointing this out. It is generally challenging to differentiate between variants of the same amino acid (for all the algorithms existing to date). In this sense, it is a limitation of our but also of all other workflows. Nonetheless, we wish to stress that we observed feature sequence divergence (using the A motif4-5 region), which helped us to separate some (but not all) of the Asp and Orn variants. For example, separations between Asp-variants are distinct (left panel in the figure below). To be on the conservative side, we only differentiated between OHAsp and Asp for our predictions, but also differentiation between DOHAsp and OHAsp would be possible. In the case of Orn-variants, there was a clear separation between Orn and the OHOrn variants (right panel). In contrast, it was difficult to differentiate between the subgroups of OHOrn variants. We believe that no A domain prediction tool will be able to solve this issue. Instead, it would be important to include information on substrate-modifying enzymes in future approaches.

**Author response image 2. sa3fig2:** 

The prediction workflow holds up well in Burkholderiales A domains, however, they fail to mention in the main text that they achieved these numbers by adding more A domains to their training set.

We thank the reviewers for this comment. We apologize for not having mentioned the training data set in the main text, while we described it in detail in the methods section (lines 714-732). We now provided more details on the analysis procedure in the main text (lines 307313). Important to note is that we did not add more A domains to the training data set but built up a new independent data set for *Burkholderiales*. The aim was to mirror the analysis we performed for pyoverdines with a completely new data set, featuring 124 A domains for training and 178 A domains as test set.

To validate their predictions, they elucidated structures of several new pyoverdines, and their predictions performed well. However, the authors did not include their MS/MS data, making it impossible to validate their structures. In general, the biggest limitation of the submitted manuscript is the near-empty methods section, which does not include any experimental details for the 20 strains or details of the annotation pipeline (such as "Phydist" and "Syndist"). The source code also does not contain the requisite information to replicate the results or re-use the pipeline, such as the antiSMASH version and required flags. That said, skimming through the source code and data (kindly provided upon request) suggests that the workflow itself is sound and a clear improvement over existing tools for pyoverdine BGC annotation.

Thank you for highlighting these issues. We agree that the methods section is short. This is because the entire paper is a step-by-step methodological introduction to our pipeline. We have now carefully revised the main text to add the information requested by the reviewer. Moreover, we have included a supplementary file with the MS/MS data of the experimentally analyzed pyoverdine structures. Finally, we further include a link to a one-click online notebook that can be used to replicate the annotation and substrate prediction results See: https://drive.google.com/drive/folders/1JsfyPUGDTFo8BDDZk8JLSvKry8emzMhr?usp=drive_ link , following a more detail explanation on code.

Predicting outer membrane receptor specificity is likewise a challenging problem and the authors have made a promising achievement by finding specific gene regions that differentiate the pyoverdine receptor FpvA from FpvB and other receptor families. Their predictions were not tested experimentally, but the finding that only predicted FpvA receptors were proximate to the biosynthesis genes lends credence to the predictive power of the workflow. The authors find predicted pyoverdine receptors across an impressive 468 genera, an exciting finding for expanding the role of pyoverdines as public goods beyond Pseudomonas. However, whether or not these receptors can recognize pyoverdines (and if so, which structures!) remains to be investigated.

Thank you for the supportive and encouraging words. The bioinformatic analysis and experimental testing of pyoverdine-receptor matching is complicated and it is not part of this paper. We treated it in a separate manuscript in which we developed an experimentally verified co-evolution algorithm that matches pyoverdines to receptors. With this algorithm, we can identify self-receptors (i.e. receptors used to take up the self-produced pyoverdine), and therefore establish pyoverdine sharing and interaction networks across strains in communities.

Please see DOI:10.1101/2023.11.05.565711 for details.

In all, the authors have assembled a rich dataset that will enable large-scale comparative genomic analyses. This dataset could be used by a variety of researchers, including those studying natural product evolution, public good eco/evo dynamics, and NRPS engineering.

Thank you for the supportive and encouraging words. We are grateful for the reviewers’ instructive suggestions and comments.

**Reviewer #3 (Public Review):**
Summary:Secondary metabolites are produced by numerous microorganisms and have important ecological functions. A major problem is that neither the function of a secondary metabolite enzyme nor the resulting metabolite can be precisely predicted from gene sequence data.In the current paper, the authors addressed this highly relevant question.The authors developed a bioinformatic pipeline to reconstruct the complete secondary metabolism pathway of pyoverdines, a class of iron-scavenging siderophores produced by Pseudomonas spp. These secondary metabolites are biosynthesized by a series of nonribosomal peptide synthetases and require a specific receptor (FpvA) for uptake. The authors combined knowledge-guided learning with phylogeny-based methods to predict with high accuracy encoding NRPSs, substrate specificity of A domains, pyoverdine derivatives, and receptors. After validation, the authors tested their pipeline with sequence data from 1664 phylogenetically distinct Pseudomonas strains and were able to determine 18,292 enzymatic A domains involved in pyoverdine synthesis, reliably predicted 97.8% of their substrates, identified 188 different pyoverdine molecule structures and 4547 FpvA receptor variants belonging to 94 distinct groups. All the results and predictions were clearly superior to predictions that are based on antiSMASH. Novel pyoverdine structures were elucidated experimentally by UHPLC-HR-MS/MS.To assess the extendibility of the pipeline, the authors chose Burkholderiales as a test case which led to the results that the pipeline consistently maintains high prediction accuracy within Burkholderiales of 83% which was higher than for antiSMASH (67%).Together, the authors concluded that supervised learning based on a few known compounds produced by species from the same genus probably outperforms generalized prediction algorithms trained on many products from a diverse set of microbes for NRPS substrate predictions. As a result, they also show that both pyoverdine and receptor diversity have been vastly underestimated.Strengths:The authors developed a very useful bioinformatic pipeline with high accuracy for secondary metabolites, at least for pyoverdines. The pipelines have several advantages compared to existing pipelines like the extensively used antiSMASH program, e.g. it can be applied to draft genomes, shows reduced erroneous gene predictions, etc. The accuracy was impressively demonstrated by the discovery of novel pyoverdines whose structures were experimentally substantiated by UHPLC-HR-MS/MS.The manuscript is very well written, and the data and the description of the generation of pipelines are easy to follow.Weaknesses:The only major comment I have is the uncertainty of whether the pipeline can be applied to more complex non-ribosomal peptides. In the current study, the authors only applied their pipeline to a very narrow field, i.e., pyoverdines of Pseudomonas and Burkholderia strains.

Thanks for your positive and encouraging comment. Regarding your only major comment, we think that the design concept of our pipeline has the potential to be applied to more complex non-ribosomal peptides. Currently, our method is tailored to accurately predict the structural composition of the Pseudomonas siderophore pyoverdine (see also response 3). A key point emphasized in our article is the importance of considering phylogeny in developing substrate prediction algorithms for A domains. Currently, the main challenge in advancing these algorithms is the limited availability of data on A domains and their corresponding substrates. However, with the future accumulation of more reference data, we are confident that the design principles of our method will enable precise predictions of the structural compositions of all products synthesized by non-ribosomal peptide synthetases (see our discussions in lines 634-645).

**Recommendations for the authors:**

**Reviewer #1 (Recommendations For The Authors):**
I believe that the manuscript would benefit from focusing solely on the task of improving pyoverdine predictions. This aspect alone is significant, and robustly supporting this claim would strengthen the manuscript. The diversity analysis provided is valuable and would undoubtedly benefit the scientific community. However, additional systematic comparisons with other methods are necessary. Furthermore, clarification of certain terms, such as 'featurebased' (e.g., whether it refers to NRPS domains or CDS), would enhance clarity.

Thank you for the supportive and encouraging words. We followed the reviewer’s suggestion and now provide the requested method comparison, see also response 2 for details. Furthermore, we have carefully checked the main text to clarify terms whenever needed. Specifically, we now define the terms “feature sequence” and “feature sequence distance” in lines 227-229.

Additionally, several minor points could be improved upon:In line 85, clarification is needed on how pyoverdine genes were identified.

Thank you for your thorough review. In the introduction section, we provided a brief overview of our work, while the detailed methodology is outlined in the results section on lines 160-174.

In line 382, it would be helpful to know the source of the sequences.

We agree and have now carefully revised the manuscript following your suggestions (lines 403-405).

Line 392 could be explained more clearly. Does it mean that the authors used an hmm search to search pHMMs against each reference sequence?

Thanks for your comment. Yes, we used an hmm search to search pHMMs against each reference sequence. We have now revised the manuscript to improve explanations (lines 413-418).

**Reviewer #2 (Recommendations For The Authors):**
The authors state they "elucidated the chemical structure of the 20 pyoverdines using culturebased methods combined with UHPLC-HR-MS/MS", so I was alarmed to see that KR and LB already published several of those structures in the cited paper. I hope that this "double dipping" will be fixed in a revision process.

Thank you for pointing this out. We agree that we have not explained clearly enough what steps were conducted in this study and which data were used from a previous paper (https://doi.org/10.1007/s00216-022-03907-w). The genomes of the 20 strains used for the verification analysis (Fig. 3d) were sequenced as part of this study (access code now provided). 14 out of the 20 pyoverdine structures were elucidated with UHPLC-HR-MS/MS in this study. For 6 out of the 20 pyoverdines, we had structural information already at hand from the previous paper. We have now clarified these details in our manuscript (lines 276-280).

Thank you for providing the source code and data, and I hope that the final non-redundant dataset will be uploaded to Zenodo or another repository. Please deposit the 20 newlysequenced genomes to GenBank or another public repository. Please also show the UHPLC-HR-MS/MS data, preferably in the form of raw data uploaded to GNPS.

We have followed the reviewer’s advice and deposited our data:

- The sequences of the 20 newly sequenced strains are available on ENA accession PRJEB76792.

- The MS/MS plots of the 14 newly analyzed pyoverdines are shown in the Supplementary Materials.

- We provide a one-click online notebook to allow readers to replicate the pyoverdine cluster annotation and substrate prediction of the 20 experimentally analyzed strains.

I suggest adding "at least" or a similar qualifier when the 73 variants are mentioned unless the literature search was truly exhaustive. What were the criteria for inclusion of the 13 strains in Table S2? For instance, sequenced strains P. syringae 1448a (10.1186/1471-2180-11-218) and P. entomophila L48 (10.1007/s10534-009-9247-y) were not included.

Thank you for your comment. We have now carefully revised the manuscript following your suggestions (lines 291-295). Regarding the criteria for including the 13 strains in Table S2, we aimed to select strains with the high credibility for inclusion in the training set data. The primary reason for excluding the two strains from the training set is that their siderophore structures were analyzed through supervised experiments. We wanted to avoid any form of biases that bioinformatic pre-predictions could introduce to downstream analyses (see Response 13 for details).

OHAsp in pyoverdines has been reported to arise from hydroxylation of Asp after it's already been activated by the A domain (10.1073/pnas.1903161116). Was there a clear difference between A domains that lead to Asp and OHAsp? Conversely, acetylation and formylation of OHOrn occur before adenylation. Can your workflow be used to differentiate cOHOrn, fOHOrn, and AcOHOrn, which are currently difficult to predict through genome mining?

Thank you for these considerations. We treated these aspects in our response 8.

Throughout, define non-proteinogenic AA substrate abbreviations (ex: Rsc, Dab).

Revised as per suggestion (lines 329-333).

Additional line comments:189: Mention PhyloPhlAn in the main text.

Revised as per suggestion (lines 189).

191: Define these filtering/selection criteria.

Thanks for your comment, we have added the criteria in the main text (line 196 and line 198).

309, 620: An A domain presumably loading histidine is present in sequenced strain P. entomophila L48 (10.1007/s10534-009-9247-y). Please also clarify that Val has previously been seen in a pyoverdine (it is in Table S1) albeit not sequenced.

We have clarified these aspects as per suggestion (lines 314-315 and line 630).

310: The pipeline can "highlight" new substrates, but not identify them.

Revised as per suggestion (line 295).

354: Please clarify "13 amino acid substrates form the core of all the 188 pyoverdine structures", considering that 279 A domain substrates couldn't be predicted.

Thanks for your comments. We have now clarified “our analysis found that 13 amino acids form the main structural substrates of all the 188 pyoverdine structures.” (lines 360-363)

630: "discovered" implies that there is experimental evidence. I suggest something like "here we predicted 151 putatively new variants".

Revised as per suggestion (line 648).

**Reviewer #3 (Recommendations For The Authors):**
Weakness:The only major comment I have is the uncertainty of whether the pipeline can be applied to more complex non-ribosomal peptides. In the current study, the authors only applied their pipeline to a very narrow field, i.e., pyoverdines of Pseudomonas and Burkholderia strains

Thanks for your comment. Please see our Responses 3+13 above, where we treat this concern in detail. Moreover, we discussed the possibility of extension to other groups of secondary metabolites in our discussion. We believe that we deliver a balanced view on the applicability of our approach and the next steps to be taken.

Please comment on this aspect.Minor:(1) When you speak about "synthesis" it is rather biosynthesis. Synthesis is chemical synthesis.Please replace all instances of the word synthesis with biosynthesis.

Revised as per suggestion.

(2) Line 188: synthetase is rather synthetases

Revised as per suggestion (line 191).